# Harmonized Learning with Concurrent Arbitration: A Brain-inspired Motion Planning Approach

## Abstract

Motion planning, regarded as a sequential decision-making problem, poses a challenge for robots in high-dimensional continuous environments due to inefficient sampling. In contrast, humans inherently possess a distinctive advantage in decision-making by leveraging limited information, primarily relying on the concurrent reasoning mechanism in the prefrontal cortex. Motivated by this, we propose a brain-inspired Deep Reinforcement Learning scheme for planning, called Harmonized Learning with Concurrent Arbitration (HLCA). The approach effectively mimics human capacity for concurrent inference tracks and the ability to harmonize strategies. Specifically, in the planning process, a general Concurrent Arbitration Module (CAM) is designed to balance the exploration-exploitation dilemma simply and efficiently. Besides, the harmonized style facilitates robots self-improving learning during the learning process, enabling the selection of appropriate strategies to guide planning. Experimental results show that HLCA outperforms the state-of-the-art benchmarks in terms of three representative metrics, which confirms the potential of emulating human-like capabilities to enhance the intelligence and efficiency of robotic planning.

## 1 Introduction

Fast and reliable motion planning in high-dimensional continuous environments is a crucial component of robot operations (Janson et al., 2015; Strub & Gammell, 2020; Fishman et al., 2023). Sampling-based methods have demonstrated impressive performance in addressing high-dimensional continuous motion planning problems. However, sampling-based methods still grapple with the exploration-exploitation trade-off dilemma (Hao et al., 2023). Traditional sampling-based methods (Kavraki et al., 1996; Triche et al., 2022), typically employ uniform sampling of the space, i.e., exploring the entire space indiscriminately to obtain new information, yet often failing to exploit the inherent structural knowledge of the problem (Hausknecht & Stone, 2015). Heuristic sampling methods (Rickert et al., 2008; Paxton et al., 2017), have been introduced to balance exploration-exploitation. However, these hand-crafted heuristics may not generalize effectively when confronted with new problems (Zhang et al., 2018). Previous work has primarily tackled this trade-off using specific techniques, such as $\varepsilon$-greedy algorithms (Rodrigues Gomes & Kowalczyk, 2009) and Upper Confidence Bound (UCB) (Garivier & Moulines, 2011), which may not always promote effective balance. Furthermore, these methods address planning problems in isolation and lack of full exploitation of prior experience and models (Kim et al., 2018). Considering these limitations, developing an intelligent and efficient exploration-exploitation trade-off mechanism is pressing.

To further enhance sampling efficiency, learning-based sampling methods have emerged as a key approach for improving motion planning performance. These approaches typically treat motion planning as a sequential decision problem (Bivard et al., 2020), which can be naturally addressed through reinforcement or imitation learning. For example, conditional variational autoencoders (Ichter et al., 2018) and motion planning networks (Qureshi et al., 2020) apply imitation learning to collected samples to guide subsequent sampling. NEXT (Chen et al., 2020) embeds a high-dimensional continuous state space into a low-dimensional representation and employs a gated path planning network (Lee et al., 2018) to predict samples. In addition, Chen et al. (Yu & Gao, 2021) leverage graph neural networks and attentional mechanisms to accelerate the search for collision-free paths. Given the

inherent diversity of motion planning problems, training samples are often generated by experts or other methods. However, current research frequently overlooks the optimization of training buffer (Ott et al., 2022). Moreover, under high-dimensional continuous environments, guiding robots towards optimal or near-optimal solutions using only a single strategy becomes particularly difficult (Osband et al., 2018). Thus, a need exists for harmonizing multi-strategy capable of efficiently adapting to complex environments.

Fortunately, owing to the gradual clarity of the decision-making capability of the brain, we can draw inspiration from its processes. In recent years, brain-inspired research has presented promising results (Radulescu et al., 2021; Xing et al., 2021; Binz & Schulz, 2022). However, these efforts have not studied the decision-making capability of the human brain nor considered harmonized learning and concurrent inference. Human decision-making is distinguished by efficient use of previous knowledge, flexible exploration based on task demands, and reduced susceptibility to external influences (Oaksford & Chater, 2009). The prefrontal cortex (PFC) plays a crucial role in human decision-making, operating on two concurrent inference tracks (Donoso et al., 2014). Besides, human working memory allows the brain to monitor and harmonize multiple strategies simultaneously (Cowan, 2005a), which enables humans to respond flexibly and efficiently to various decision-making scenarios.

Inspired by the concurrent reasoning mechanism in the PFC, we propose **Harmonized Learning with Concurrent Arbitration (HLCA)**, a brain-inspired Deep Reinforcement Learning (DRL) algorithm for motion planning. This work contains a general Concurrent Arbitration Module (CAM) and a Harmonized Self-Improving Learning (HSIL) style. We demonstrate the effectiveness of our work in high-dimensional continuous maze planning tasks. HLCA outperforms the state-of-the-art in three different evaluation metrics, which indicates the potential of human-like learning to enhance the capabilities of intelligent machines. We release our code and data in the supplementary material. Our main contributions can be summarized as follows.

- The novel incorporation of the concurrent reasoning mechanism introduces innovative enhancements to traditional decision-making models.

- When sampling during the planning process, the CAM employs a dynamic switching mechanism to simultaneously evaluate exploration and exploitation options, effectively balancing the binary choice dilemma.

- Throughout the training process, the HSIL style allows robots to improve themselves from historical quality experience. HLCA mitigates the risk of falling into local optimal that often results from relying on a single strategy.

## 2 PREFRONTAL FOUNDATIONS OF HUMAN REASONING

Theoretically, human decision-making can approximate Bayesian reasoning, effectively utilizing limited information to make properly qualified decisions (Collins & Koechlin, 2012). This process is intricately linked to the concurrent reasoning mechanism in PFC (Donoso et al., 2014; Cohen et al., 2007; Domenech et al., 2020). The current experimental evidence from neuroscience using brain scans suggests that the concurrent reasoning mechanism in PFC includes two concurrent inference tracks (Donoso et al., 2014), shown in Figure 1. Specifically, the first medial track consists of ventromedial PFC (vmPFC), pregenual anterior cingulate cortex (pgACC), dorsomedial PFC (dmPFC), and the striatum. Within this track, the vmPFC-pgACC performs reliability inference on the ongoing strategy. When the ongoing strategy becomes unreliable, the dmPFC triggers exploration by creating a probing strategy. Subsequently, the striatum confirms the new strategy once it becomes reliable. The second lateral track involves the frontopolar cortex (FPC) and middle lateral prefrontal cortex (mid-LPC). The FPC can infer the reliability of alternative strategies, and the mid-LPC exploits the reliable alternative strategy and rejects the unreliable probing strategy.

Furthermore, the human brain exhibits a remarkable ability in this process is also related to monitoring and harmonizing multiple strategies (Donoso et al., 2014; Cowan, 2005a). However, due to limitations in working memory capacity (Cowan, 2005b), the current experimental evidence from neuroscience suggests that the most accurate decisions are made when the number of monitored strategies is limited to four (Cowan, 2001). The brain mechanism of human decision-making reasoning upon which we draw in this study is confined to the outcomes that preceded our investigation.

## 3 PRELIMINARIES

### 3.1 SETTING FOR PLANNING

Assume given a planning problem $U_i = (\mathcal{S}, \mathcal{S}_w, \mathcal{S}_f, s_s, s_g)$, where $\mathcal{S}$ is the state space of the problem, i.e., the configuration of robots and its operation workspace, $\mathcal{S}_w$ is workspace operated by robots, $\mathcal{S}_f = \mathcal{S} \setminus \mathcal{S}_o$ is the free space, $\mathcal{S}_o$ is the obstacles set. $s_s, s_g \in \mathcal{S}$ are the start state and goal state, respectively. The purpose of planning problems is to find a collision-free path $\xi$ in free space $\mathcal{S}_f$ from the start state $s_s$ to the goal state $s_g$. Tree Sampling Algorithms (TSA) start with the $s_s$ as the root node of the search tree $\mathcal{T}$. The search tree $\mathcal{T}$ is progressively ***expanded*** by sampling new states in $\mathcal{S}$, connecting these sampled states until a leaf node of $\mathcal{T}$ reaches $s_g$. A path $\xi = \{[s_{i-1}, s_i]\}_{i=1}^{T}$ can be extracted based on the search tree $\mathcal{T}$, where $[s_{i-1}, s_i] \subset \mathcal{S}$ with $s_0 = s_s$ and $s_T = s_g$. If these path segments are detected by the collision detection module to be free of obstacles, $\xi$ is a collision-free path. Let $\sum_{i=1}^{T} c(\{s_{i-1}, s_i\})$ denotes the path cost of the collision-free path $\xi$.

### 3.2 NOTATIONS

In our work, the strategy set is denoted by $\mathbf{\Omega} = \{\omega_1, \omega_2, \omega_3, \omega_u\}$, which is similar to the long-term memory. The inference buffer $\mathcal{I} = \{\omega_1, \omega_2, \omega_u\}$, which is designed by imitating the inference buffer of the concurrent reasoning model in PFC (Donoso et al., 2014). $\omega_u$ is a learning-based planner, whereas $\omega_1, \omega_2, \omega_3$ are are non-learning planners. The initialization of $\mathbf{\Omega}$ and $\mathcal{I}$ are manually designed. At episode(problem) $t$, given $\boldsymbol{h_t}$, the reliability of $\omega_i$ in $\mathcal{I}$ involves two distinct phases. See section 4.3.1 for a detailed description. $\boldsymbol{h_t}$ is all possible histories before $t$-episodes, i.e., $\boldsymbol{h_t} = ([U_0, \mathcal{T}_0, c_0], \dots, [U_{t-1}, \mathcal{T}_{t-1}, c_{t-1}])$, which are sampled from the training buffer $\mathcal{D}$.

## 4 METHOD

### 4.1 MODELING REASONING PROCESS IN THE HUMAN PFC

A theoretical model of human decision-making reasoning is proposed to describe the concurrent reasoning mechanism in the PFC, shown in Figure 1. We hypothesize that the inference buffer $\mathcal{I}$ is limited to three or four (Collins & Koechlin, 2012), which contains the ongoing strategy $\omega_k$ and other alternative strategies $\omega_i, \omega_j$. The reliability of each monitored strategy in $\mathcal{I}$ i.e., $\lambda_i, \lambda_j, \lambda_k$ is inferred via Bayesian reasoning. The reasoning process can be summarized as follows.

- If $\lambda_k$ is reliable, the ongoing strategy selects actions with maximal rewards.

- When the reliability of all strategies in $\mathcal{I}$ becomes unreliable, the robot would switch to the exploration and then develop a probing strategy from long-term memory.

- The robot would return to exploitation in two cases: (i) any of the monitored strategies in $\mathcal{I}$ becomes reliable, this monitored strategy is exploited and the probing strategy is rejected. (ii) The probing strategy is reliable while other monitored strategies are unreliable, and then the probing strategy is confirmed by storing it in $\mathcal{I}$.

Moreover, if $\mathcal{I}$ reaches its maximal capacity, the strategy recently used is discarded from the inference buffer, while it is still stored in long-term memory.

### 4.2 NEURAL GUIDED CONCURRENT ARBITRATION

Motivated by concurrent inference tracks in the concurrent reasoning model (Donoso et al., 2014), we propose a Concurrent Arbitration Module (CAM), shown in Figure 2. This module leverages a neural-constrained observation function that incorporates priors, enabling simultaneous exploration and exploitation. Thus, robots can explore new possibilities and advance to expansion more effectively. Suppose that $\omega_u$ has the value function $V(s; \boldsymbol{\theta})$ and policy function $\pi(a|s; \boldsymbol{\theta})$[1], which are

---

[1]In the motion planning setting, $s'$ is the cascaded state from current state $s$ after taking action $a$.

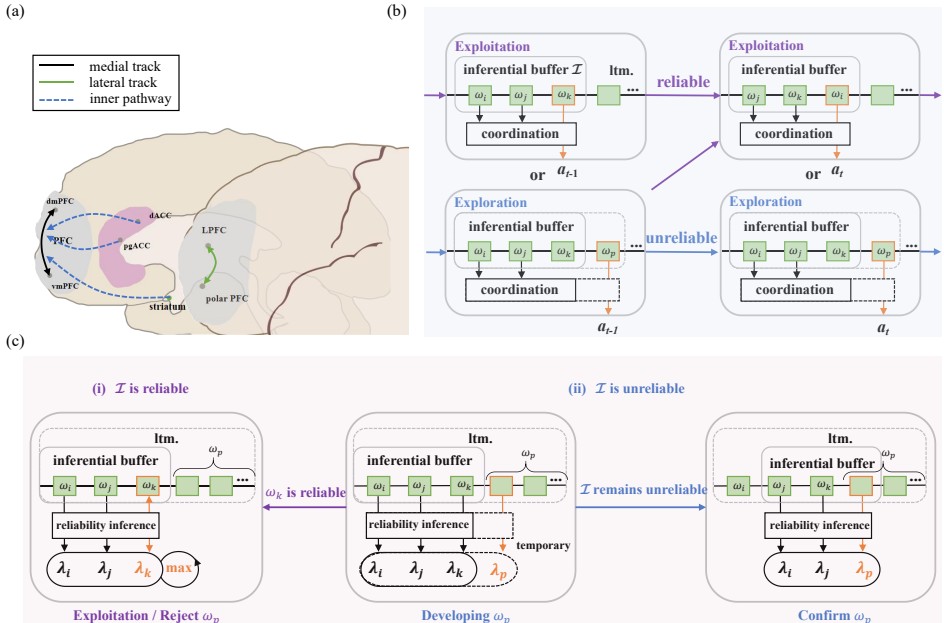

Figure 1: Concurrent reasoning. (a) Concurrent reasoning mechanism in PFC. (b) Modeling concurrent inference tracks. (c) Modeling multi-strategies harmonization. Green cubes represent existing strategies stored in long-term memory (ltm), and the orange cube is the ongoing strategy. When all strategies in $\mathcal{I}$ are unreliable(blue), the robot switches to exploration and develops a probing strategy $\omega_p$ from ltm. The exploration phase continues until the monitored strategy(purple) or a probing strategy $\omega_p$ is reliable(blue). Then $\omega_p$ would be rejected (red) or confirmed (yellow) subsequently.

merged into a two-headed neural network with shared parameters $\boldsymbol{\theta}$. The former assesses the current state as either good or bad, and the latter helps robots select actions.

Robots select a node as parent state $s_p$ from existing tree $\mathcal{T}$ and expand the new state $s_{new}$ in the neighborhood $s_p$ as an infinite-armed bandit problem. Candidate states $\mathcal{S}_l^c$ are sampled guided by the policy network $\pi(s'|s; \boldsymbol{\theta})$, $\mathcal{S}_l^c = \left\{ s_1^c, \ldots, s_{N_c}^c \right\}$. As the algorithm proceeds, the number of states gradually increases and their values become correlated, the traditional UCB algorithm is not directly applicable to balance exploration and exploitation. To overcome this issue, we explicitly model these correlations (Chen et al., 2020), which facilitates the exploration of unknown spaces.

$$\sigma_l(s) = \sqrt{\frac{\log \sum_{s' \in \mathcal{S}} \kappa(s')}{\kappa(s)}}, \quad \text{with } \kappa(s) = \sum_{s' \in \mathcal{S}} k\left(s', s\right) \tag{1}$$

where $\sigma_l(s)$ is the variance estimator after $l$-times expansion. Kernel smoothing is used for $\sigma_l(s)$ and $k\left(s', s\right)$ is the kernel function. In addition, the generalized structure of problems can be captured more accurately by leveraging priors and environmental information. In particular, the final expansion will be selected from candidates via the observation function $\phi(s)$, which is given by

$$\phi(s) = (1 - \varepsilon) \cdot \text{softmax}\left(\bar{r}_l(s) + c \cdot \sigma_l(s)\right) + \varepsilon \cdot c([s', s_g]) \tag{2}$$

where the noise parameter $\varepsilon$ and $c$ are constants less than 1. $\bar{r}_l$ is the average reward after kernel smoothing, i.e., $\bar{r}_l(s) = \frac{\sum_{s' \in \mathcal{S}} k\left(s', s\right) r\left(s'\right)}{\sum_{s' \in \mathcal{S}} k(s', s)}$, $r(s)$ can be replaced by $V(s; \boldsymbol{\theta})$.

If the maximum observation of candidate states is better than a threshold $\beta$, then the candidate state with the maximum observation will be expanded as $s_{new}$. On the contrary, if all observations are below $\beta$, the robot will be triggered to explore new possibilities. For this purpose, temporary probed states $\mathcal{S}_l^p = \left\{ s_1^p, \ldots, s_{N_c}^p \right\}$ is guided by the policy of other strategies in $\mathcal{I}$. There are two scenarios

in which the robot switches to exploitation, (i) when all exploration rounds are completed and no eligible state is discovered, the robot will stop exploring and choose the state with the highest value of $\phi(s)$ among all states that have been found so far. (ii) When the maximum observation of probed states exceeds $\beta$, then this state will be expanded as $s_{new}$ and the robot switches to exploitation.

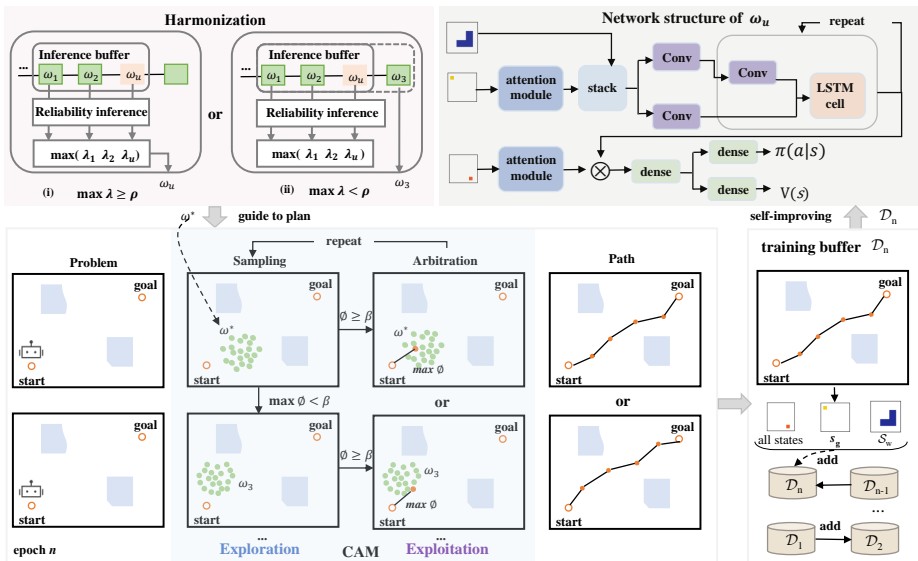

Figure 2: Overall model architecture. For each problem, the harmonization module performs reliability inference for the strategies (green cubes) in $\mathcal{I}$ and then selects $\omega^*$. Then $\omega^*$ guides the expansion by concurrently estimating the observations for exploration and exploitation based on the CAM. These historical paths are stored in the $\mathcal{D}$, which is used to self-improve the $\omega_u$ (the pink cube). This interactive learning is more beneficial for buffer collection.

### 4.3 HARMONIZED SELF-IMPROVING LEARNING

In this section, we introduce a novel Harmonized Self-Improving Learning (HSIL) approach, which can optimize the collection of the training buffer. $V(s; \boldsymbol{\theta})$ and $\pi(s'|s; \boldsymbol{\theta})$ can be improved based on prior successes. Before solving each problem, a desirable strategy among the inference buffer has the highest reliability. Besides, the HLCA can effectively guide the planning in the test phase. The training buffer $\mathcal{D}$ is a collection of solutions for planning path.

#### 4.3.1 RELIABILITY INFERENCE

The first phase occurs before planning, i.e., the ex-ante reliability $\lambda_{i,t}$. The second phase takes place after planning, i.e., the ex-post reliability $\mu_{i,t}$. $\lambda_{i,t}$ reflects the confidence level assigned to each strategy in $\mathcal{I}$ before their execution. It provides an assessment of the expected reliability of the strategies in guiding the robot's planning process. In contrast, $\mu_{i,t}$ is determined by evaluating the actual outcomes observed after execution. It takes into account the feedback received from the environment and provides a measure of the reliability of the strategies based on their actual performance. This dual estimation process enhances the robot's ability to dynamically adjust its strategies. The ex-ante and ex-post reliabilities are respectively expressed as:

$$\lambda_{i,t} = \frac{F(\omega \mid U_t, \mathcal{I}) \left[ (1-\tau)\mu_{i,t-1} + \tau \sum_{j \in \{1,\ldots,N_{\mathcal{I}}\}} \mu_{j,t-1} \right]}{Z_t^\lambda}, \tag{3}$$

$$\text{with } \mu_{i,t} = \frac{\mathcal{P}_{\omega_i}(s' \mid s) \cdot (f_{colli} + \sum c([s, s']))}{Z_t^\mu} \tag{4}$$

where $F(\omega \mid U_t, \mathcal{I})$ is a probability distribution function, which encodes the likelihood of selecting each strategy in $\mathcal{I}$. The parameter $\tau$ is perceived volatility, $Z_t^\mu$ and $Z_t^\lambda$ are regularization factors.

$\mathcal{P}_{\omega_i}(s' \mid s)$ is a state transition function following a strategy $\omega_i$. $f_{colli}$ and $\sum c([s, s']$ are collision checks and path costs after standardization, respectively. Assume that the selection of any strategy in $\mathcal{I}$ has an equal likelihood of occurring. Thus, $F(\omega \mid U_t, \mathcal{I})$ does not require calculation and can be directly used as a constant.

### 4.3.2 SELF-IMPROVING LEARNING

The reliability inference function can harmonize multiple strategies to guide the planning process, progressively optimizing the collection of the training buffer $\mathcal{D}$. The value function $V(s; \boldsymbol{\theta})$ and policy function $\pi(s'|s; \boldsymbol{\theta})$ can be learned as $\mathcal{D}$ expands. To be clear, in $n$-th epoch, the training buffer is denoted by $\mathcal{D}_n$. In the initialization epoch, the $\omega_u$ is not prioritized for reliability inference as $V(s; \boldsymbol{\theta})$ and $\pi(s'|s; \boldsymbol{\theta})$ are poorly trained. Hence, the local optimal strategy $\omega^*$ is selected from $\omega_1, \omega_2$ according to Equation 3. Then, the path generated by $\omega^*$ is added to $\mathcal{D}_0$ for each planning problem. Once $m$ planning problems are solved, $\mathcal{D}_1$ will be used to train $V(s; \boldsymbol{\theta})$ and $\pi(s'|s; \boldsymbol{\theta})$.

In the $n$-th $(n \neq 0)$ training epoch, $\omega_u$ will participate in reliability inference. If the reliability of the most reliable strategy in $\mathcal{I}$ is higher than the predefined threshold $\rho$, it is selected as an optimal strategy $\omega^*$. The solution generated by the guidance of $\omega^*$ is added to $\mathcal{D}_n$ until $m$ planning problems are solved. Then $\mathcal{D}_n$ serves as the training sample to train $V(s; \boldsymbol{\theta})$ and $\pi(s'|s; \boldsymbol{\theta})$ according to the objective function which is given by

$$\ell_n = \sum_{i=1}^{n \cdot m} (V(s_i; \boldsymbol{\theta}) - v_i)^2 - \sum_{i=1}^{n \cdot m} \log \pi(s_{i+1}|s_i; \boldsymbol{\theta}) + \alpha \cdot \|\boldsymbol{\theta}\|_2^2 \tag{5}$$

where the value $v_i$ is estimated from empirical paths in $\mathcal{D}_n$, i.e., $v_i = \sum_i^T c([s_i, s_{i+1}])$. $\alpha$ is a regular term. The parameters for $V(s; \boldsymbol{\theta})$ and $\pi(s'|s; \boldsymbol{\theta})$ are learned while the model continues to accumulate experience. $V(s; \boldsymbol{\theta})$ and $\pi(s'|s; \boldsymbol{\theta})$ gradually learn from previous successes and improve themselves as they encounter more problems and better solutions. This incremental expansion of the training buffer allows for self-improving learning and adaptation of the model.

On the contrary, if the reliability of all strategies in $\mathcal{I}$ is below the threshold for several consecutive times, a new strategy $\omega_3$ would be chosen from $\Omega$. Then the solution generated by the guidance of $\omega_3$ is added to $\mathcal{D}_n$. The entire procedure is repeated in an epoch until all problems are solved.

## 5 EXPERIMENTS

### 5.1 GENERAL SETTINGS

**Environment Datasets.**   We conduct a series of experiments to evaluate the performance of our work in solving high-dimensional continuous maze planning tasks. Our experiments involve eight distinct datasets [2] shown in Figure 3 and describe each of them in detail as follows.

(a) **Maze2** contains Easy2 and Hard2, which involves a 2-degree of freedom (DoF) point robot operating in a 2D workspace. The obstacle density of the Hard2 is set to a minimum of 46%, ensuring that the distance from the start state to the goal is no less than 1.

(b) **Maze3** involves a 3-DoF stick robot operating in a 2D workspace.

(c) **UR5** comprises a 6-DoF robot operating in a 3D workspace. Some randomly generated objects, namely two sets of boxes, poles, and pads, are generated in two different size ranges.

(d) **Snake7** consists of a 7-DoF snake robot operating in a 2D workspace. The snake robot has five sticks and its end position is 2-DOF. The 2D workspace is the same as Maze2D.

(e) **Kuka7** features a 7-DoF robot operating in a 3D workspace. The robot is in a fixed base position. In Kuka7, Kuka13, and Kuka14, all boxes are random in the workspace for each problem.

(f) **Kuka13** comprises a 13-DoF robot operating in a 3D workspace.

---

[2] https://github.com/rainorangelemon/gnn-motion-planning/tree/main/maze_files

(g) **Kuka14** has a robot with two 7-DoF Kuka arms operating in a 3D workspace. Each arm must successfully achieve its goal region while avoiding any potential collisions with its environment.

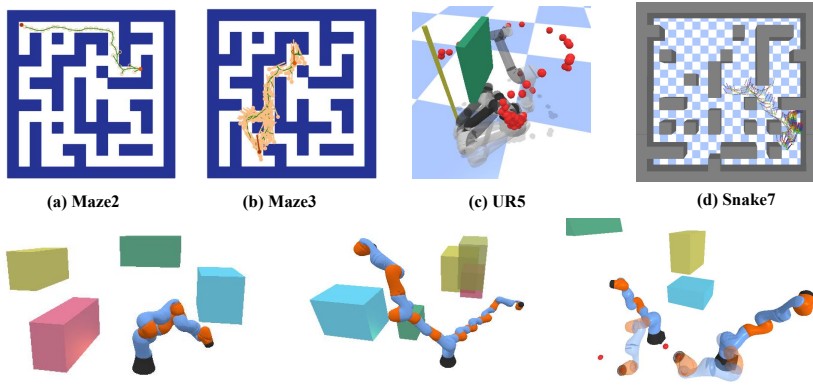

Figure 3: All environment datasets. The top line from left to right: (a) Maze2; (b) Maze3; (c) UR5; (d) Snake7. The bottom line from left to right: (e) Kuka7; (f) Kuka13; (g) Kuka14.

**Baselines.** We select some motion planning methods known for their superior performance. These methods include RRT* (Karaman & Frazzoli, 2011) with a target bias heuristic, informed-RRT* (Gammell et al., 2014) with an informed search strategy, and the state-of-the-art batch-sampling approach BIT* (Gammell et al., 2015). We also incorporate LazySP (Haghtalab et al., 2018), which leverages manually designed heuristics to decrease the number of collision checks without relying on specific problem information. Furthermore, we include NEXT (Chen et al., 2020), which integrates neural priors into the reward function to boost sampling efficiency based on DRL. GNN and GNN+smoother (Yu & Gao, 2021) employ graph neural networks (GNNs) for path exploration and smoothing, thereby reducing collision checks.

**Settings.** There are 3000 different problems in each environment, with 2000 for training and 1000 for testing. For each one, the environment is randomly configured. The start and goal states are sampled uniformly from $\mathcal{S}_f$, and the workspace is generated randomly based on a fixed distribution.

## 5.2 OVERALL PERFORMANCE

To provide a thorough evaluation of our method, we perform multiple tests in each environment leveraging 1000 test problems. The maximum number of samples for all algorithms is set to 1000 for consistency. We assess the performance with four key metrics. Namely, the success rate in finding collision-free paths, the path cost, the collision checks, and the total running time. The success rate represents the percentage of collision-free paths out of the total paths. The path cost is determined by summing the Euclidean path length for each problem. The collision checks metric is the number of times the planning process detects whether the robot collides with an obstacle or not. The total running time is planning time to run 1000 test problems. Quantitative evaluation and detailed results analysis are presented in the Appendix A.

Based on the record of the results presented in Figure 4, our method HLCA exhibits remarkable performance across all test environments, achieving a 100% success rate. Notably, the HLCA significantly reduces the number of collision checks, particularly in high-dimensional environments. Although HLCA may not always achieve optimal path costs in all environments, our results consistently demonstrate near-optimal performance. Moreover, in the case of high-dimensional environments ranging from Kuka7 to Kuka14, HLCA exhibits minimum path cost. The results of the visualization are in the Appendix F.

## 5.3 INFERENCE BUFFER CAPABILITY

Similar to human working memory, we hypothesize that there is a constraint on the working capacity of the inference buffer in our method HLCA. To demonstrate this hypothesis, we select five different

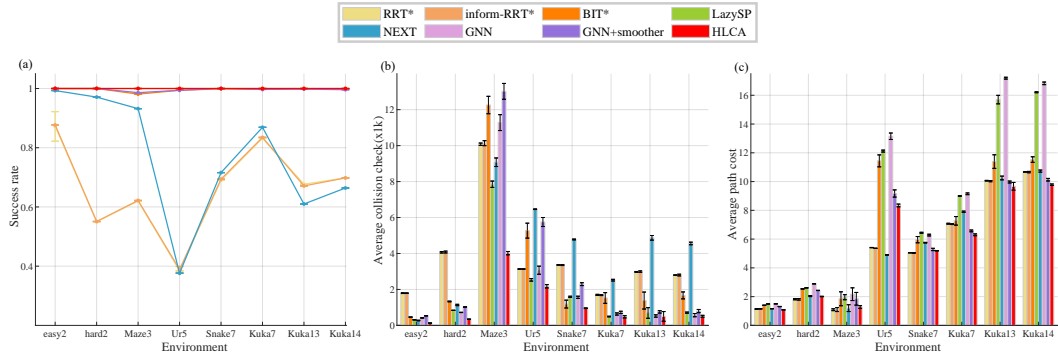

Figure 4: Performance comparison in all environments. From left to right: (a) Success rate. (b) Average collision checks. (c) Average path cost.

strategies, i.e., RRT*, BIT*, LazySP, NEXT, and GNN. The capacity of the inference buffer $\mathcal{I}$ is set among $N_{\mathcal{I}} = \{2, 3, 4, 5\}$. Then we conduct experiments in five different environments, i.e., Hard2, UR5, Kuka7, Kuka13, and Kuka14. The experimental results are shown in Figure 5.

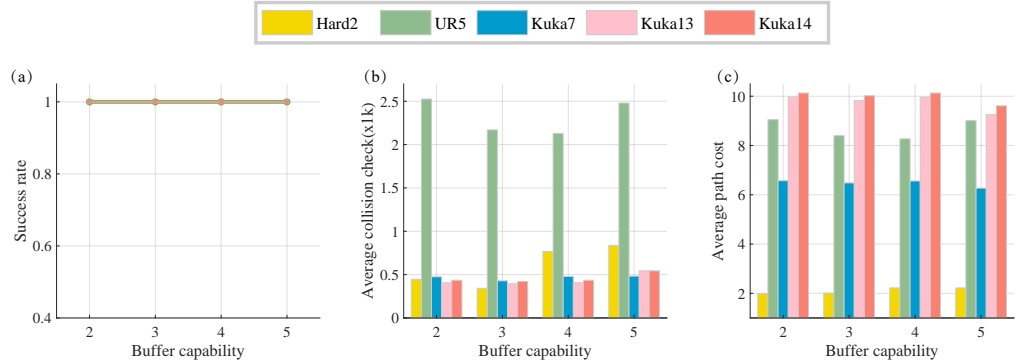

Figure 5: Performance comparison with different buffer capability settings. (a) Success rate. (b) Average collision checks. (c) Average path cost.

The best performances in terms of the number of collision checks and path costs are achieved when the capacity of $\mathcal{I}$ is set to 3 or 4. Meanwhile, we observe a decline in the overall performance across all five environments when the capacity increases to 5. The results suggest that $\mathcal{I}$ also has a capacity limit, which may be related to the reliability of the inference function designed to imitate human brain reasoning. Moreover, the complexity of the environment affects the buffer capacity requirements. Specifically, our experiments demonstrate that the best performance is achieved in relatively simple environments, such as Hard2, when the buffer capacity is set to 3. While in more complex environments, such as UR5, the optimal buffer capacity is up to 4. It is important to note that even when the buffer capacity remains constant, the choice of strategies within the buffer can significantly impact the final results.

## 5.4 Ablation Study

**Ablation study I: concurrent guidance expansion.** To validate the effectiveness of the CAM, we integrate it into the NEXT algorithm. The remaining components of the NEXT algorithm are unchanged. The results shown in Figure 6 demonstrate the effectiveness of the CAM in finding better paths with fewer collision checks and higher success rates across all environments. Moreover, the CAM optimizes the path cost in high-dimensional environments.

**Ablation study II: harmonized self-improving learning.** To demonstrate the efficiency of HSIL, the inference buffer $\mathcal{I} = \{RRT^*, BIT^*, NEXT\}$, where NEXT serves as $\omega_u$. These strategies in $\mathcal{I}$

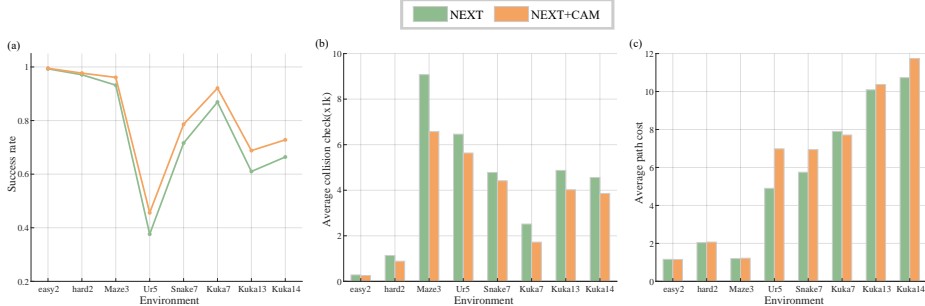

Figure 6: Performance comparison for the CAM. From left to right: (a) Success rate. (b) Average collision checks. (c) Average path cost.

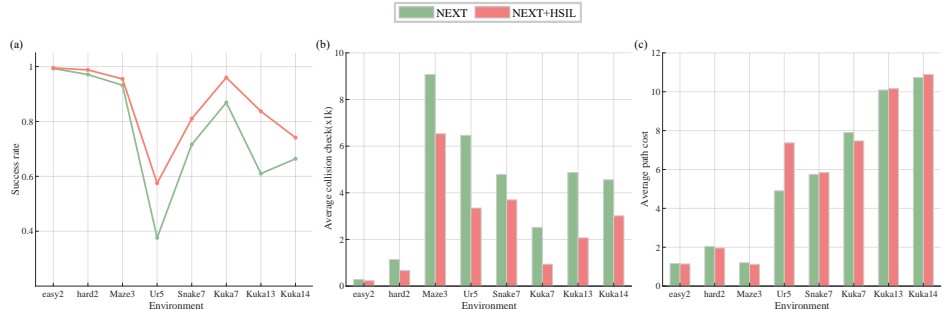

Figure 7: Performance comparison for HSIL. From left to right: (a) Success rate. (b) Average collision checks. (c) Average path cost.

are inferred through reliability $\lambda$. Harmonization is only used to optimize buffer collection during the learning process. The results are presented in Figure 7, which demonstrate the effectiveness of HSIL in finding collision-free paths with fewer collision checks and higher success rates across all environments.

**Ablation study III: varying the maximum sampling number.** We perform experiments and discuss the influence of the maximum sampling number in our methods. Detailed experimental results and analysis are seen in the Appendix B.

## 6 CONCLUSION

In this paper, we present a brain-inspired DRL approach, Harmonized Learning with Concurrent Arbitration (HLCA), inspired by the concurrent reasoning mechanism in the human PFC to enhance planning performance in high-dimensional continuous spaces. Our method incorporates a general Concurrent Arbitration Module (CAM) to address the exploration-exploitation dilemma. With the proposed HLCA, the limitation of traditional single-strategy methods can be overcome. Experiment results demonstrate the robustness and stability of HLCA in complex environments. Yet, we only focus on motion planning in high-dimensional continuous spaces in this work. For future research, we will extend the HLCA to other complex decision-making scenarios to evaluate its effectiveness and adaptability. Furthermore, exploring the decision-making mechanisms in the human brain can potentially contribute to the development of more advanced and reliable robotic systems, such as emulating human cognitive capabilities.

AUTHOR CONTRIBUTIONS

ACKNOWLEDGMENTS

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

# A    MORE RESULTS AND PERFORMANCE ANALYSIS

In this section, we quantitatively evaluate the overall performances of all the methods across various environments. Each performance metric contains average values and corresponding standard deviations, which provide insights into the consistency and variability of the results.

Table 1: Success rate. In all environments, HLCA is able to find a collision-free path with a success rate of 100%.

| | Easy2 | Hard2 | Maze3 | UR5 | Snake7 | Kuka7 | Kuka13 | Kuka14 |
|---|---|---|---|---|---|---|---|---|
| HLCA | **1.00± 0.00** | **1.00± 0.00** | **1.00± 0.00** | **1.00± 0.00** | **1.00± 0.00** | **1.00± 0.00** | **1.00± 0.00** | **1.00± 0.00** |
| GNN+smoother | **1.00± 0.00** | **1.00± 0.00** | 0.99± 0.00 | 0.96± 0.00 | 0.99± 0.00 | 0.99± 0.00 | 0.99± 0.00 | 0.99± 0.00 |
| GNN | **1.00± 0.00** | **1.00± 0.00** | 0.99± 0.00 | 0.96± 0.00 | 0.99± 0.00 | 0.99± 0.00 | 0.99± 0.00 | 0.99± 0.00 |
| NEXT | 0.99± 0.00 | 0.97± 0.00 | 0.93± 0.00 | 0.38± 0.00 | 0.72± 0.01 | 0.87± 0.01 | 0.61± 0.01 | 0.66± 0.00 |
| BIT* | **1.00± 0.00** | **1.00± 0.00** | 0.98± 0.00 | 0.99± 0.00 | **1.00± 0.00** | 0.99± 0.00 | 0.99± 0.00 | 0.99± 0.00 |
| LazySP | **1.00± 0.00** | **1.00± 0.00** | 0.99± 0.00 | 0.99± 0.00 | **1.00± 0.00** | **1.00± 0.00** | **1.00± 0.00** | **1.00± 0.00** |
| informted-RRT* | 0.88± 0.00 | 0.56 ± 0.00 | 0.62± 0.00 | 0.40± 0.01 | 0.69± 0.00 | 0.83± 0.00 | 0.67± 0.00 | 0.70± 0.00 |
| RRT* | 0.87± 0.00 | 0.55 ± 0.00 | 0.62± 0.00 | 0.39± 0.00 | 0.69± 0.00 | 0.84± 0.00 | 0.67± 0.01 | 0.68± 0.00 |

**Success rate.**    The results presented in Table 1 highlight the performance of different methods in terms of success rate in finding collision-free paths across various test environments. Our proposed method HLCA achieves a remarkable 100% success rate in all the test environments, indicating its high reliability and effectiveness. By comparison, RRT* and informed-RRT*, which employ random extensions with target bias, exhibit lower success rates in certain environments. For instance, RRT* achieves an 87% success rate in the Easy2 environment, and this rate drops to 55% in the low-dimensional complex Hard2 environment. The meta-RL-based method NEXT performs well in low-dimensional spaces but struggles in high-dimensional complex environments, with a success rate of 37% in the UR5 environment. The state-of-the-art batch-sampling method BIT* consistently achieves a success rate of no less than 99% across all environments. In addition, employing GNN to accelerate motion planning algorithms in batch-sampling methods outperforms BIT*. LazySP also performs well (not less than 99%) in all environments. To sum up, our method outperforms or performs competitively with existing methods across a range of environments, emphasizing its effectiveness and superiority in finding collision-free paths.

Table 2: Average collision checks. HLCA is able to achieve the best performance in all environments.

| | Easy2 | Hard2 | Maze3 | UR5 | Snake7 | Kuka7 | Kuka13 | Kuka14 |
|---|---|---|---|---|---|---|---|---|
| HLCA | **129.05± 3.04** | **342.35± 13.90** | **4019.10± 92.49** | **2175.48± 94.14** | **955.05± 17.66** | **479.86± 63.64** | **500.43± 269.34** | **508.03±52.56** |
| GNN+smoother | 533.71± 9.41 | 1023.16± 8.71 | 13018.36± 440.52 | 5750.87± 250.91 | 2294.44± 79.87 | 725.30± 61.47 | 751.52±66.97 | 797.20± 89.22 |
| GNN | 416.54± 9.23 | 720.47± 7.46 | 11280.04± 442.34 | 3069.73± 225.07 | 1562.94± 60.27 | 633.79± 60.39 | 522.87±64.25 | 570.51± 90.29 |
| NEXT | 272.28± 14.81 | 1135.6± 37.7 | 9074.94± 237.82 | 6461.4± 16.47 | 4779.74± 22.73 | 2516.75 ± 45.73 | 4870.57± 129.12 | 4557.94 ± 71.22 |
| BIT* | 460.24± 5.55 | 1331.02±19.16 | 12258.05± 484.30 | 5270.79± 415.24 | 1185.82± 218.00 | 1523.72± 297.33 | 1382.40±471.56 | 1668.67 ± 192.56 |
| LazySP | 309.64± 4.27 | 836.80± 8.34 | 7851.77± 176.01 | 2533.09± 68.50 | 1588.88± 34.37 | 1492.77± 17.43 | 691.37±295.54 | 709.22± 32.34 |
| imformted-RRT* | 1791.55± 15.26 | 4080.78 ± 51.50 | 10134.94± 137.76 | 3134.25± 10.70 | 3355.42± 26.53 | 1685.87 ± 26.92 | 2991.66±41.12 | 2801.03±47.42 |
| RRT* | 1793.62± 21.49 | 4065.63 ± 40.78 | 10078.18± 65.00 | 3135.39± 9.94 | 3362.65± 14.35 | 1695.87± 31.82 | 2975.61±23.41 | 2798.55± 9.56 |

**Average collision checks.**    In Table 2, our method HLCA demonstrates superior performance than existing methods in terms of collision checks across all tested environments. RRT* and informed-RRT* require a higher number of collision checks compared with other methods across all tested environments. NEXT performs well in the Easy2 environment with only 272.28 collision checks, but it still faces challenges with redundant sampling in higher dimensions, particularly in the UR5 and Kuka14 environments. LazySP and GNN can significantly reduce collision checks by employing hand-crafted heuristics and designing path explorers that iteratively predict collision-free edges, respectively. Notably, HLCA requires only 28%, 25%, 32%, 42%, 93%, 18%, 34%, and 27% of the collision checks performed by BIT*. GNN requires 322%, 210%, 283%, 141%, 164%, 132%, 104%, and 112% of the collision checks performed by HLCA in various environments. In Kuka14, only 11% of the collision checks performed by NEXT are required by HLCA. All these results indicate that HLCA is highly efficient in finding collision-free paths.

Table 3: Average path cost. HLCA can achieve a relatively low path cost in all environments, except for the UR5 environment. Moreover, it achieves the lowest path cost from Kuka7 to Kuka14.

| | Easy2 | Hard2 | Maze3 | UR5 | Snake7 | Kuka7 | Kuka13 | Kuka14 |
|---|---|---|---|---|---|---|---|---|
| HLCA | 1.08± 0.00 | 2.01± 0.00 | 1.28± 0.01 | 8.34± 0.04 | 5.19± 0.02 | **6.30± 0.03** | **9.67± 0.04** | **9.78± 0.09** |
| GNN+smoother | 1.31± 0.00 | 2.43± 0.01 | 1.85± 0.00 | 9.17± 0.19 | 5.28± 0.03 | 6.57± 0.05 | 9.97± 0.10 | 10.13± 0.05 |
| GNN | 1.50± 0.00 | 2.89± 0.01 | 2.17± 0.02 | 13.15± 0.28 | 6.28± 0.03 | 9.15± 0.07 | 17.19± 0.23 | 16.83± 0.26 |
| NEXT | **1.06± 0.00** | 2.04± 0.00 | 1.20± 0.00 | **4.90± 0.04** | 5.75± 0.02 | 7.90± 0.05 | 10.25± 0.11 | 10.73± 0.09 |
| BIT* | 1.40± 0.00 | 2.54± 0.00 | 1.85± 0.00 | 11.44± 0.14 | 5.96± 0.03 | 8.27± 0.03 | 11.39± 0.03 | 11.53 ± 0.02 |
| LazySP | 1.49± 0.00 | 2.61± 0.00 | 1.96± 0.00 | 12.12± 0.19 | 6.44± 0.02 | 9.00± 0.04 | 15.70± 0.11 | 16.21± 0.17 |
| imformted-RRT* | 1.15± 0.00 | **1.81 ± 0.00** | 1.10± 0.00 | 5.36± 0.06 | **5.04± 0.06** | 7.04± 0.06 | 10.03± 0.05 | 10.66± 0.01 |
| RRT* | 1.14± 0.00 | 1.82 ± 0.00 | **1.09± 0.00** | 5.41± 0.04 | 5.05± 0.04 | 7.07± 0.01 | 10.06± 0.05 | 9.56 ± 0.01 |

**Average path cost.**    Table 3 presents the average path cost of different methods that successfully find collision-free paths. Significantly, our method HLCA demonstrates exceptional performance in terms of path cost. It achieves very low path costs in all environments and outperforms other methods in Kuka7 to Kuka14. The results highlight the effectiveness of HLCA in finding collision-free paths with low path costs, making it a promising approach for motion planning in complex environments. Despite lower success rates and higher numbers of collisions, RRT* and informed-RRT demonstrate low average path costs in all environments. The results demonstrate the lowest path cost of 1.81, 1.09, and 5.04 in Hard2, Maze3, and Snake7 environments, respectively. This can be attributed to the way their target bias is sampled, resulting in relatively simple solution paths and low path costs. The GNN+smoother method, which incorporates path smoother and barrier encoding, achieves superior performance in high-dimensional environments compared with other baselines. However, LazySP, which relies on manually designed heuristics that do not prioritize path cost reduction, exhibits much higher path costs in all environments.

Table 4: Total running time.

| | Easy2 | Hard2 | Maze3 | UR5 | Snake7 | Kuka7 | Kuka13 | Kuka14 |
|---|---|---|---|---|---|---|---|---|
| HLCA | 5.18 ± 0.04 | 6.86± 0.11 | 5.64 ± 0.13 | 17.18± 0.45 | 10.20± 0.39 | 14.13± 0.53 | 9.52± 0.44 | 10.78± 0.42 |
| GNN+smoother | 1.99± 0.11 | 3.05± 0.05 | 7.40± 0.42 | 2.60± 0.33 | 1.26± 0.32 | 5.03± 1.86 | 1.50 ± 0.54 | 1.22± 0.31 |
| GNN | 1.70± 0.10 | 2.86± 0.05 | 7.16 ± 0.42 | 1.98± 0.32 | 1.15± 0.32 | 4.85± 1.86 | 1.35± 0.54 | 1.11± 0.31 |
| NEXT | 3.17± 0.04 | 5.13± 0.09 | 2.62± 0.04 | 23.49± 0.06 | 16.67± 0.25 | 18.57± 1.00 | 13.02± 0.03 | 15.89± 0.13 |
| BIT* | **0.47± 0.01** | **0.73± 0.02** | 2.01± 0.15 | 1.44± 0.44 | **0.36± 0.05** | 2.25± 1.09 | 0.98± 0.44 | **0.83 ± 0.43** |
| LazySP | 0.84± 0.03 | 1.48± 0.02 | 4.22± 0.12 | 2.27± 0.17 | **0.36± 0.02** | 1.93± 0.01 | **0.69 ± 0.30** | 0.99± 0.16 |
| imformted-RRT* | 1.02± 0.01 | 1.03 ± 0.02 | 1.01± 0.02 | 1.02± 0.02 | 1.01± 0.01 | **0.98± 0.02** | 0.88± 0.01 | 1.01± 0.16 |
| RRT* | 1.00± 0.01 | 1.00± 0.01 | **1.00± 0.01** | **1.00± 0.00** | **1.00± 0.00** | 1.00± 0.03 | 1.00± 0.01 | 1.00± 0.01 |

**Total running time.**    Table 4 demonstrated the total running time for 1000 test problems for all algorithms to find a successful path. A commonly raised issue regarding learning-based algorithms is the time overhead attributed to the frequent utilization of a large neural network model during solving problems, such as NEXT. The running time is higher than non-learning algorithms because our algorithm is improved based on NEXT and runs multiple candidate strategies concurrently. Nevertheless, our algorithm reduces the running time compared to NEXT. Future work could focus on using pre-computation or caching, e.g., reusing past results to overcome some of this overhead.

## B  ABLATION STUDY:VARYING THE MAXIMUM SAMPLING NUMBER

In this section, we investigate the impact of the maximum sampling number $N_T$ on CAM and HLCA, respectively.

**CAM.**    We vary the Maximum sampling number$N_T$ among the values $\{200, 500, 1000, 2000, 4000, 6000\}$ and analyze the corresponding impact on three metrics. The results are presented in Figure 8. The results indicate that a higher value of $N_T$ leads to a higher success rate. This demonstrates the significance of $N_T$ in achieving better performance in planning problems. As the maximum sampling number increases, the robot can explore more

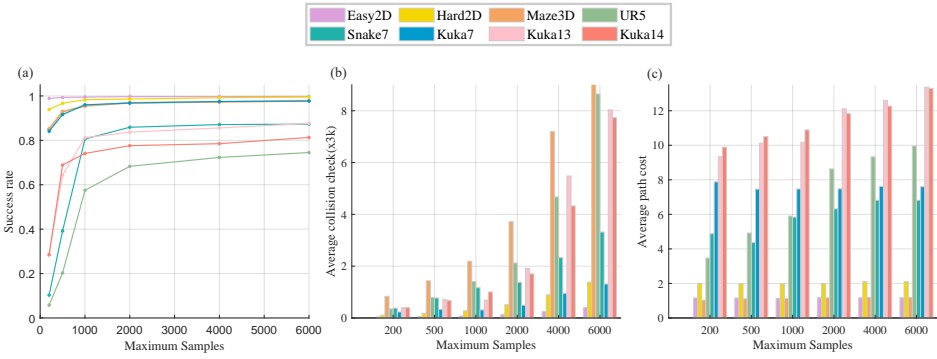

Figure 8: Performance comparison for CAM with different maximum sampling number. From left to right: (a) Success rate. (b) Average collision checks. (c) Average path cost.

state spaces and collect more information, which in turn improves the success rate. However, it is important that increasing the value of $N_T$ also results in a higher number of collision checks, which can affect the efficiency of planning. Therefore, there is a trade-off between samples and efficiency. In certain environments, such as UR5, setting a specific value for $N_T$, i.e., $N_T = 1000$, is necessary to achieve a good success rate while maintaining a reasonable level of efficiency. Interestingly, the path cost remains almost constant for all settings of $N_T$ within the given range, which indicates that the model is robust to changes in the maximum sampling number and that the path cost is not significantly affected by variations in $N_T$.

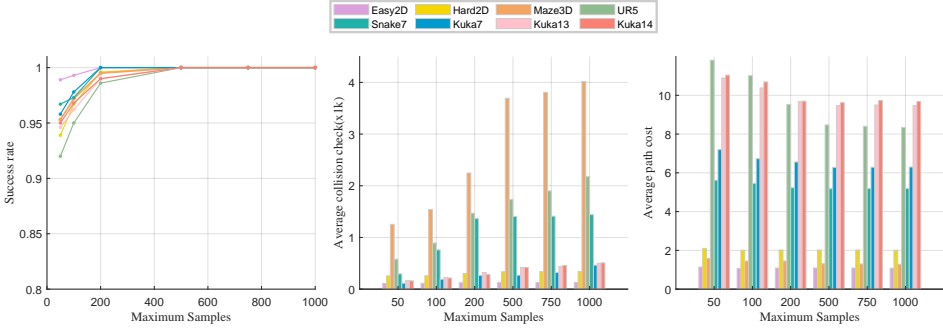

Figure 9: Performance comparison for HLCA with different maximum sampling number. From left to right: (a) Success rate. (b) Average collision checks. (c) Average path cost.

**HLCA.** We vary the maximum sampling number $N_T$ among $\{50, 100, 200, 500, 750, 1000, 2000, 4000, 6000\}$ on HLCA. To show more clearly the change in performance before reaching the plateau, the results before 1000 was shown in Figure 9. The results indicate that performing consistently across different settings of the maximum sampling number $N_T$. HLCA reaches a plateau in all environments when $N_T$ is 500 in success rate and path cost. When $N_T$ is 1000, it achieves the highest success rate and exhibits robustness in various scenarios, which highlights the effectiveness and adaptability in planning.

The computation cost on the GPU is measured and presented in Figure 10. As the maximum sampling number ($N_T$) increases, the robot needs to perform collision checking on states, resulting in additional computational time. The results highlight the trade-off between performance and computational cost. It is worth noting that beyond a certain point, increasing the maximum sampling number does not yield significant improvements in the success rate but only leads to an increase in the computational resources required. Therefore, it becomes crucial to determine the optimal value for the maximum sampling number that strikes a balance between achieving desirable performance and managing computational costs effectively.

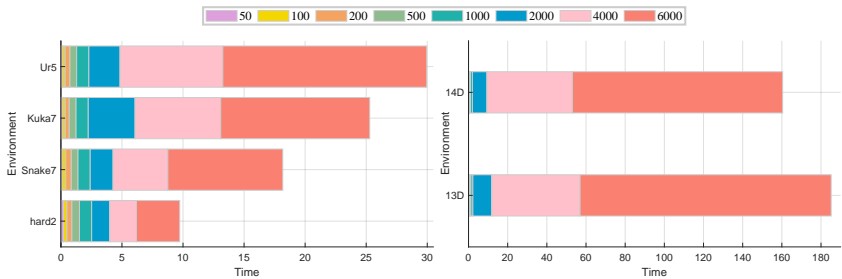

Figure 10: Computation cost in different environments with different sample limits. The runtime for each environment is normalized based on the result when the maximum sample number is 1000.

## C  HYPERPARAMETERS

These hyperparameters play a crucial role in determining the efficiency of the planning process. Here, key hyperparameters are shown in Table C.

Table 5: Hyperparameter settings

| Hyperparameters | Values |
|---|---|
| $\alpha$ | 0.7 |
| $\beta$ | 0.9 |
| Learning rate | 1e-3 |
| Training iteration | 20 |
| added $m$ for each epoch | 200 |
| $k$-NN for GNN | 20 |
| Training batch size | 8 |
| Step size in 2D, 3D | 5e-2 |
| Step size in others | 5e-1 |
| Maximum sample number | 1000 |
| Batch size for GNN, BIT*, LazySP | 100 |

## D  MORE CHALLENGING ENVIRONMENTS

We generate two groups of new test problems in the 2D environment, i.e., Dense2 and Large2. The environment settings are as follows:(i) Dense2: size, $15 \times 15$, obstacle density, no less than 0.60. (ii) Large2: size, $50 \times 50$, obstacle density, no less than 0.50. For Hard2, size is $15 \times 15$, obstacle density is not less than 0.46. Maintaining consistent parameters across the environments, we tested for all methods in new problems. The results are shown in D.

Table 6: Results in the more challenging environments

| | Dense2 | | | | Large2 | | | |
|---|---|---|---|---|---|---|---|---|
| | success rate | collision checks | path cost | running time | success rate | collision checks | path cost | running time |
| HLCA | **0.99 ± 0.01** | **1959.04± 22.88** | 2.56± 0.02 | 12.18± 0.55 | 0.05± 0.00 | 3256.82± 10.25 | 2.52± 0.04 | 68.53± 1.26 |
| GNN+smoother | 0.96± 0.01 | 2727.04± 43.06 | 3.25± 0.02 | 5.65± 0.12 | 0.02± 0.00 | 5135.97± 30.76 | 2.54± 0.02 | 378.33± 0.31 |
| GNN | 0.96± 0.01 | 2304.78± 41.51 | 3.96 ± 0.02 | 5.45± 0.12 | 0.02± 0.00 | 5135.39± 31.31 | 2.82± 0.08 | 378.31± 3.38 |
| NEXT | 0.82± 0.01 | 5032.77± 50.21 | 2.38± 0.02 | 15.02± 1.89 | 0.00± 0.00 | / | / | 85.80± 1.28 |
| BIT* | 0.95± 0.03 | 2628.65± 22.39 | 3.44± 0.01 | 1.39± 0.03 | 0.05± 0.00 | 9843.90± 129.08 | 3.22± 0.47 | 9.80± 0.22 |
| LazySP | **0.99± 0.01** | 2001.21± 19.35 | 3.44± 0.01 | 3.54± 0.11 | 0.02± 0.00 | 3162.30± 15.08 | **2.12 ± 0.11** | 93.39± 1.12 |
| informted-RRT* | 0.19± 0.01 | 4911.51± 20.36 | 1.80± 0.04 | 1.07 ± 0.02 | **0.69 ± 0.01** | **2836.94± 61.29** | 10.42± 0.03 | 3.67± 0.07 |
| RRT* | 0.19± 0.01 | 4929.20± 30.44 | **1.79± 0.04** | **1.00± 0.00** | 0.00± 0.00 | / | / | **1.00± 0.03** |

Compared to Hard2, operating in the Dense2 environment with a high obstacle density demands increased costs, collision checks, and runtime to reach the goal state. Our method performs best in terms of success rate and collision checks. While RRT* and informed-RRT* demonstrate a low

---

**Algorithm 1** HLCA

---

**Input**:$V(s; \boldsymbol{\theta})$, $\pi(a|s; \boldsymbol{\theta})$
**Parameter**:batchsize $N_b$, training set size $m$

1: Initialize $\mathcal{D}_0 \leftarrow \emptyset$
2: **for** epoch $n = 1, 2, \cdots, N$ **do**
3:    **while** $i < m$ **do**
4:       Sample a planning problem $U_i$
5:       Obtain local optimal strategy $\omega_i^*$ and path $\xi$
6:       $\mathcal{D}_n \leftarrow \mathcal{D}_n \cup (\xi, U_i)$ and $i \leftarrow i + 1$
7:    **end while**
8:    **for** $j = 1, 2, \cdots, n * m$ **do**
9:       $d \leftarrow \emptyset$
10:      **repeat**
11:        Sample $d_j$ from $\mathcal{D}_n$
12:        $d \leftarrow d \cup d_j$
13:      **until** $len(d) > N_b$
14:      $\theta \leftarrow \theta - \frac{\eta}{N_b} \nabla_\theta \sum_{i=1}^{N_b} \ell_n(V, \pi; d)$
15:    **end for**
16: **end for**

---

**Algorithm 2** Reliability Inference

---

**Input**:$\alpha$, $U_i$, $\mathcal{I}$, $\boldsymbol{\Omega}$
**Output**:local optimal strategy $\omega_i^*$

1:
2: $\omega_i \leftarrow argmax_{\omega \in \mathcal{I}} \lambda_\omega$ and obtain $\lambda_{max}$
3: **if** $\lambda_{max} \geq \alpha$ **then**
4:    $\omega_i^* = \omega_i$
5: **else**
6:    Sample probe strategy $\omega_p \sim \boldsymbol{\Omega} \setminus \mathcal{I}$
7:    $\omega_i^* = \omega_p$
8: **end if**

---

path cost, their success rate lags behind other methods, possibly attributed to their efficacy in solving easy problems. NEXT exhibits lower path costs compared to other learning-based methods, but the collision checks are nearly twice as high.

For large environments, all algorithms demonstrate a notable reduction in success rate. Only a few instances where the distance between the start state and the goal state is not long succeed in finding a path. The majority of failures may be attributed to the limitation of the maximum number of samples, set as 1000. Particularly noteworthy is the superior performance of informed-RRT*, because informed sampling limits the sampling area and reduces the need for samples. For future work, we will try to set a higher maximum number of samples and introduce informed sampling to enhance the performance of our algorithm in large environments.

## E ALGORITHM

The pseudocodes of all our algorithms are provided in 1, 2, and 3.

## F VISUALIZATION

We illustrate the path generated by RRT*, NEXT, and HLCA with 1000 samples in Figure 11, Figure 12 on Hard2 and Maze3 environments, respectively.

---

**Algorithm 3** CAM

---

**Input**: problem $U$, threshold $\beta$, exploration limit $T_{explor}$, probe strategy $\omega_p$, local optimal strategy $\omega^*$

**Output**: $s_{new}$

  1:  Initialize $\xi \leftarrow \emptyset$, $\Phi \leftarrow \emptyset$, $S \leftarrow \emptyset$
  2:  **for all** episode $t = 0, 1, \cdots, T$ **do**
  3:      Sample candidate states guided by $\omega^*$ $\mathcal{S}_l^c \sim \mathcal{S}$
  4:      $s' \leftarrow argmax_{s \in \mathcal{S}_l^c} \phi(s))$ and obtain $\phi_{max}$
  5:      **if** $\phi_{max} \geq \beta$ **then**
  6:          $s_{new} = s'$
  7:      **else**
  8:          $\Phi \leftarrow \Phi \cup \phi_{max}$ and $S \leftarrow S \cup s'$
  9:          **while** $\phi_{max} < \beta$ and $t \leq T_{explor}$ **do**
 10:              Sample probe states guided by $\omega_p$ $\mathcal{S}_l^p \sim \mathcal{S}$
 11:              $s' \leftarrow argmax_{s \in \mathcal{S}_l^p} \phi(s)$ and obtain $\phi_{max}$
 12:              $\Phi \leftarrow \Phi \cup \phi_{max}$ and $S \leftarrow S \cup s'$
 13:              $t \leftarrow t + 1$
 14:          **end while**
 15:          Obain index of $\max \Phi$ $i$ and $s_{new} \leftarrow S_i$
 16:      **end if**
 17:  **end for**
 18:  **return** $s_{new}$

---

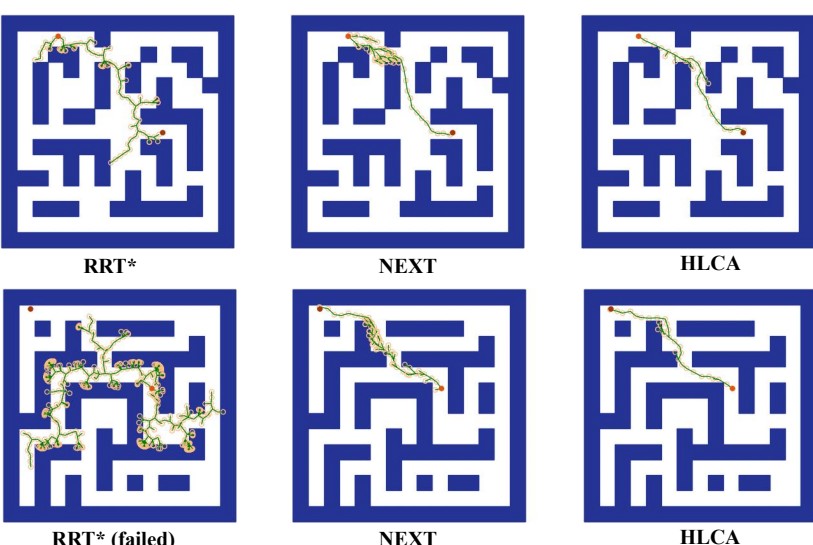

Figure 11: Planned paths in Hard2. In the top line, the collision checks of RRT*, NEXT, and HLCA are 2.56k, 1.80k, and 1.60k and the path costs of RRT*, NEXT, and HLCA are 1.40, 1.37, and 1.32. In comparison to RRT* and NEXT, the HLCA finds a successful path with significantly fewer collisions and lower path costs. In the bottom line, the difference between the HLCA algorithm and RRT* is particularly noticeable. The HLCA finds an optimal collision-free path, while RRT* fails to do so. The collision checks of NEXT and HLCA are 1.92k, 1.78k and the path costs of NEXT and HLCA are 1.36, 1.27. The HLCA demonstrates how it could improve planning effectiveness in complex environments.

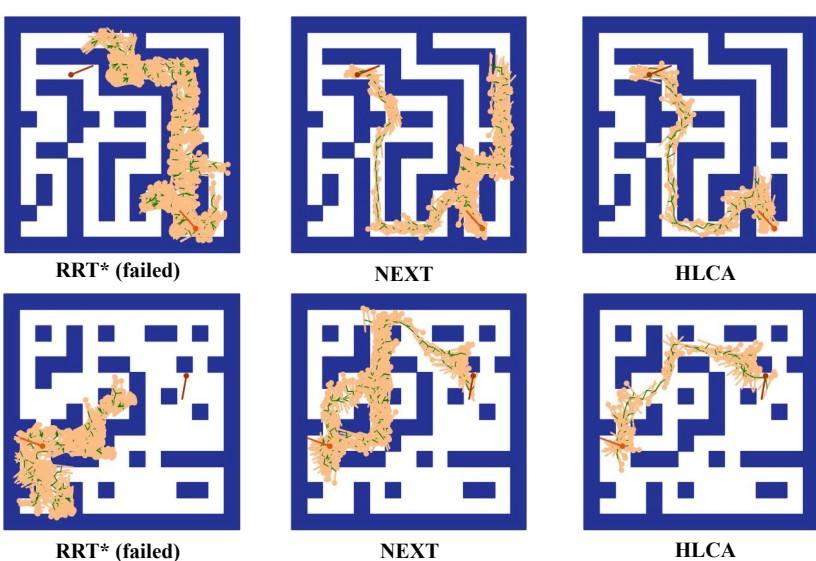

Figure 12: Planned paths in Maze3. In high-dimensional environments, the planning performance of RRT* experiences a notable decline. In the top line, the collision checks of NEXT and HLCA are 49.71k, 12.23k and the path costs of NEXT and HLCA are 2.78, 2.74. In the bottom line, the collision checks of NEXT and HLCA are 10.08k, 6.93k and the path costs of NEXT and HLCA are 2.55, 2.39. The HLCA outperforms both RRT* and NEXT in terms of collision checks and path costs. This result demonstrates that our method significantly enhances planning performance in high-dimensional complex environments. The HLCA addresses the limitations of RRT* and provides a more efficient solution.

