# OpenReview forum: "Harmonized Learning with Concurrent Arbitration: A Brain-inspired Motion Planning Approach"
_ICLR.cc/2024/Conference — ICLR 2024 Conference Withdrawn Submission_

### Official Review · Reviewer_azvY · 2023-10-30

**Soundness:** 1 poor
**Presentation:** 1 poor
**Contribution:** 2 fair
**Rating:** 5
**Confidence:** 4

**Summary:**

In this work the authors draw inspiration from the human biological brain to design an algorithm capable of adaptively switching from exploitation to exploration, amongst multiple strategies for path planning. Drawing inspiration from the function of the human Prefrontal Cortex (PFC), the authors have developed a model that can select amongst multiple stored planning strategies, augmenting their performance by utilizing space exploration when the observed reliability of the known strategies is compromised.  At the core of the authors work lies the Concurrent Arbitration Module(CAM), which is responsible for assessing the performance of the currently followed strategy and deciding whether to keep following it, switch to another one or explore new possibilities by sampling from the environment.
At each planning iteration, the agent rolls out new trajectories starting from a starting state, attempting to reach a goal state. The agent selects, at each step, the action that leads to the state with the maximal predicted reward and also features a reliability exceedinga hyperparameter beta. If the current strategy omega_k cannot produce new states that have an estimated reliability above beta, then the strategies in the Inference buffer I are considered. If these strategies cannot also produce reliable states, then the agent switches to a new strategy stored in the long term memory buffer U and uses it to explore new state possibilities. If the new probing strategy omega_p can be considered reliable, then it is chosen otherwise the agent simply adopts the strategy with the highest reliability and continues roll outs. The agent adds rolled out paths to a training buffer D, and every 200 paths it updates its value and policy estimators, which are given by a network with shared parameters. The agent utilizes several metrics to produce the reliability measures of each strategy, such as probability of collision from each state, cost  and estimated reward. It uses a bootstrapping method where confirmed values ( after trajectory executions) are used to update the reliability estimator.

**Strengths:**

The main contribution of this work, the ability to switch between strategies and alternate between exploration and exploitation when the agent is uncertain about next steps, is predicated on a traditional learning objective, that updates the value and policy network parameters after observing past path choices.

The method is founded upon a solid idea of attempting to gauge the reliability of multiple strategies each time the agent finds itself in uncertain situations. The agent can utilize the diverse set of outcomes from the multiple of strategies to escape local optima that can lead to unimprovable performance.

The authors use traditional metrics such as path cost, estimated reward and collision probability to attempt to gauge a path's feasibility in a convincingly feasible manner.
They use post state expansion feedback to improve reliability for each strategy.

The method seems to be able to have offline training capacity by updating past example's reliability score through equation 3.

**Weaknesses:**

The author’s method requires a few critical hyperparameters which might be difficult to predefine in complex problems. The reliability threshold, being the most critical, appears to be extremely sensitive to not only the environment but also subtasks with an environment. Additionally, the perceived volatility parameter ti, is rather arcane in how it is selected.

The following assumption is given without adequate proof or reasoning.
 “Since π(s ′ |s; θ) is trained based on prior local optimal successes on various challenges, it can lead the policy to gradually approach the optimal policy.”
Why does the superimposition of locally optimal solutions approach an overall optimal policy?
While it might make intuitive sense, and can have practical soundness in several cases.

Paper has several typos and requires additional proofreading to correct them i.e
Section 4 line 2 “which can optimize the collection of training buffer and V (s; θ) and π(s ′ |s; θ) can be self-improving learned” or Section 4.3.1 last line of last paragraph “Thus, F(ω | Ut, I) does not require calculation and can be direct as a constant”, to name a few.

No mention of the resources and time complexity the method requires versus the competition.

An algorithmic presentation of the algorithm is mandatory. The writing style made it rather hard to understand what is learned, when the strategies are rotated etc.

The authors mention that 2 ⁄ 3 strategies are handcrafted. How are they thus and how do they operate? Since the authors do have a learnable strategy trainable by backpropagation on their loss signal, why wouldn't they accommodate more classes or learnable policy / value networks and switch between them? This raises the question of the amount of engineering done in the hand-crafted strategies and how much they can generate states that can dislodge the learned strategy from local optima. While the multistrategy idea is quite interesting, its efficacy would be much more strongly showcased if there were multiple learned strategies or if hand crafted were indeed required, if they were simple heuristics.

**Questions:**

It is confusing when the reliability checks happen for all strategies in I. Does a complete reliability assessment of all strategies happen when the current strategy k becomes unreliable or at each state expansion?

When a current strategy  omega_k (learned) becomes inactive, and a new probing strategy is selected, how does this work? Is the new strategy omega_p initialized from omega_k and then further trained? Or are there several trainable strategies during initialization and at each time a learnable strategy omega_k becomes unreliable, a new omega_p supplants it and is trained differently? The latter would be mean that during the initial cycles more than one untrained strategy are selected and trained. Is that correct?

---

> ### Author Response · Authors · 2023-11-20
> **Response to Reviewer azvY 1/2**
>
> Thanks for your careful review and important suggestions. We have revised manuscript accordingly to make it clearer. We address the specific questions as follows.
>
> 1. Parameters
>
> The values for the reliability threshold $\alpha$ and the perceived volatility parameter $\tau$ lie within the range of $[0, 1]$. Theoretically, a larger threshold $\alpha$ leads to better results. However, setting a higher threshold will result in longer experiment runtime, with marginal or possibly no performance improvement. We adopt non-stringent thresholds, with $\alpha$ specifically set to 0.7.
>
> Given that different strategies are adaptable for dealing with distinct environments, we introduce the parameter $\tau$ to represent the preference for strategies in the inference buffer. In our experiments, we assume an equal likelihood for the selection of all strategies. Therefore, $\tau$ is set as 0.5. Besides, we provide details about the hyperparameters of the algorithm in Appendix C of the updated manuscript.
>
> 2. Optimization Problems
>
> Sorry for the inappropriate expression. We can't guarantee that $V(s;\boldsymbol{\theta})$ is an overall optimal policy. $V(s;\boldsymbol{\theta})$ can be improved based on successful experiences generated by historical local optimal strategies. Therefore, we have changed it to “$V(s;\boldsymbol{\theta})$ and $\pi(s'|s;\boldsymbol{\theta})$ gradually learn from previous successes and improve themselves as they encounter more problems and better solutions.” in the updated version.
>
> 3. Typos in writing
>
> We have improved the listed typos in the updated manuscript.
>
> 4. Resources and time complexity
>
> We calculated Params for learning-based algorithms and demonstrated the planning runtime for all algorithms to find a successful path.
>
>  |Methods|Params|
>  |:----:|:----:|
> |HLCA |0.16M |
> |GNN+smoother |0.31M|
> |GNN| 0.09M|
> |NEXT|0.16M|
>
> We demonstrated the total running time for 1000 test problems across all algorithms to find a successful path. A snapshot of the results is as follows. The complete table and discussion are updated in  Appendix A of the revised manuscript. To be more clearly represented in the figure, the results are normalized based on the running time of RRT*. HLCA, GNN+smoother, GNN, and NEXT are learning-based algorithms, others are non-learning algorithms.
>
>  |$ $|Easy2|Hard2|Maze3|UR5|Kuka7|Kuka14|
>  |:----:|:----:|:----:|:----:|:----:|:----:|:----:|
> |HLCA |3.78 $\pm$ 0.04|5.86$\pm$ 0.11|3.64 $\pm$0.13|15.18$\pm$ 0.45|12.13$\pm$ 0.53 |10.78$\pm$ 0.42|
> |GNN | 1.70$\pm$0.10|2.86$\pm$ 0.05|7.16 $\pm$ 0.42 |1.98$\pm$ 0.32|4.85$\pm$ 1.86|1.11$\pm$ 0.31|
> |NEXT| 3.17$\pm$ 0.04|5.13$\pm$ 0.09|2.62$\pm$ 0.04 |23.49$\pm$ 0.06|18.57$\pm$ 1.00|15.89$\pm$ 0.13|
> |BIT*|0.47$\pm$ 0.01|0.73$\pm$ 0.02|2.01$\pm$ 0.15|1.44$\pm$ 0.44|2.25$\pm$ 1.09|0.83 $\pm$ 0.43|
> |LazySP|0.84$\pm$ 0.03|1.48$\pm$ 0.02|4.22$\pm$ 0.12 |2.27$\pm$ 0.17|1.93$\pm$ 0.01 |0.99$\pm$ 0.16|
> | RRT*|1.00$\pm$ 0.01 |1.00$\pm$ 0.01|1.00$\pm$ 0.01 |1.00$\pm$ 0.00|1.00$\pm$ 0.03|1.00$\pm$ 0.01|
>
> A common concern about learning-based algorithms is that their time cost due to the frequent calling of a large neural network model at the inference phase, such as NEXT. The running time is higher than non-learning algorithms because our algorithm is improved based on NEXT and runs multiple candidate strategies concurrently. Nevertheless, our algorithm reduces the running time compared with NEXT. Future work could focus on using pre-computation or caching, e.g., reusing past results to overcome some of this overhead.
>
> 5. Pseudocode of the algorithm
>
> We have provided the pseudocodes of all our algorithms in Appendix E of the updated manuscript.
>
> 6. Doubts about manually designed strategies
>
> There might be some misunderstandings about our harmonized learning settings. It is the strategy set $\boldsymbol{\Omega}$ and the inference buffer $\mathcal{I}$ instead of the strategies that are manually designed, i.e., $\boldsymbol{\Omega}$ = {$\omega_{1},\omega_{2}, \omega_{3},\omega_{u}$}, $\mathcal{I}$={$\omega_{1},\omega_{2}, \omega_{u}$}, where $\omega_{1},\omega_{2}, \omega_{3}$ are existing strategies, the network structure of $\omega_{u}$ is existing. In our experiments, $\omega_{1},\omega_{2}, \omega_{3}$ indicate RRT*, BIT*, LazySP.

---

> ### Author Response · Authors · 2023-11-20
> **Response to Reviewer azvY 2/2**
>
> 7. Why not accommodate more strategies
>
> (1) more strategies
>
> Reliability inference and harmonization in the inference buffer can be time-consuming and computationally expensive. If satisfactory performance can be attained with a small number of strategies, there is no need to include additional strategies. This is particularly relevant in practical scenarios where agents must make rapid decisions.
>
> (2) more learnable policy/value networks
>
>  Thanks for your suggestion. Accommodating more learnable policy/value networks and switching between them is very valuable and we will study it in the future. The main reason that we only include a pair of policy/value networks in this experiment is as follows. The initial performance of learning-based algorithms is relatively poor and thus learning data needs to be collected by using non-learning strategies. The model gradually is self-improved upon its prior successful experience.
>
> 8. When to conduct a reliability assessment
>
> We perform reliability inference for each planning problem, instead of before expanding states.
>
> 9. Concerns about new probe strategies
>
> (1) Brain model
>
> In the human decision-making reasoning model, $\omega$ denotes a human behavioral strategy. When $\omega_{k}$ proves unreliable while other strategies in the inference buffer $\mathcal{I}$ remain reliable, the strategy with the highest reliability in $\mathcal{I}$ is selected. If none of the strategies in the inference buffer are unreliable, a new probe strategy $\omega_{p}$ is randomly developed from long-term memory. Notably, $\omega_{p}$ is not trained from the initial $\omega_{k}$ initialization.
>
> (2) Algorithm
>
> During the initialization phase, $\boldsymbol{\Omega}$ = {$\omega_{1},\omega_{2}, \omega_{3},\omega_{u}$} and $\mathcal{I}$ ={$\omega_{1},\omega_{2}, \omega_{u}$} were designed to imitate long-term memory and inference buffer, respectively, where $\omega_{1},\omega_{2}, \omega_{3}$ are existing strategies, the network structure of $\omega_{u}$ is existing. Similarly, If strategies in $\mathcal{I}$ are reliable, the strategy with the highest reliability is selected. In contrast, $\omega_{3}$ serves as a tentative probe strategy to guide agents in planning a path. This path is added to the training buffer as before. Moreover, if the strategies in $\mathcal{I}$ exhibit unreliability consecutively twice, $\omega_{3}$ replaces the strategy with the minimum reliability. Importantly, even in the scenario where $\omega_{u}$ is replaced, i.e., absent from the inference buffer, the samples collected persist in training $\omega_{u}$.
>
> (3) Experiment
>
> Observations from the experiments highlight that: two consecutive strategies in the inference buffer remain unreliable is a rare phenomenon. There are three reasons for this happen. (i) Our selected strategy demonstrates greater robustness. (ii) The threshold for reliability is not quite stringent. (iii) It is worth considering that the environments may not be quite complex.
>
> For the third reason, we validated all methods by generating environments with larger dimensions and a higher density of obstacles. Experiment details can be found in Appendix D of the updated paper.

---

### Official Review · Reviewer_Api9 · 2023-10-30

**Soundness:** 3 good
**Presentation:** 3 good
**Contribution:** 3 good
**Rating:** 6
**Confidence:** 4

**Summary:**

The author propose a motion planning algorithm, for high-dimensional continuous environments, based on two ideas inspired from models of decision making in the human brain, and the prefrontal cortex (PFC) in particular. The first idea is concurrent arbitration module (CAM), inspired by the concurrent inference track of the human PFC, allowing the algorithm to consider multiple candidates during the exploration phase using an observation function $\phi$, before deciding to switch to the exploitation phase. The second idea is to define a self-improving learning algorithm that takes into account the feedback received from the environment to improve the decision-making process by incorporating estimates of both the ex-ante and ex-post reliabilities. The learning algorithm combines a learning based planner $\omega_u$ with a number of non-learning planners $w_i$. The total number of planners in the inference buffer is limited, again, inspired by the observed number of concurrent plans in human studies. The authors evaluate the algorithm in comparison to 7 baselines over 7 different environments of varying complexity and number of DoFs.

**Strengths:**

- Extensive comparison against multiple baselines across different environments of varying complexity.
- Significantly improves over state-of-the-art across a number of key performance measures, i.e., success rate, number of collision-checks, and average path cost.

**Weaknesses:**

- To me the derivation of the main algorithm was cluttered by discussion of (models of) brain function that I wasn't able to appreciate, more so as it was difficult for me to connect it to the description of the algorithm (which itself seemed to be presented rather loosely, relying on the figures and a sequence of paragraphs tending to different aspects, issues, and proposed solutions.) It would have helped to present the main logical steps in a pseudocode listing, with pointers to equations or section numbers that deal with finer issues.
- I have a concern about basing the evaluation on the quality of the output, without examining the performance of the algorithm itself; see below.
- I also have concerns about the chosen benchmark leading the proposed algorithm to achieve 100% success rate. It is necessary to include a more challenging benchmark that specifically exposes failure cases, and compare to other baselines in that case too.

**Questions:**

**Technical comments:**
- Experiments:
    - It is necessary to include the computation/running-time of the motion planning algorithm itself, rather than just the number of collision-checks (also storage requirements for the memory buffers). Namely, as the proposed algorithm runs multiple candidate strategies in parallel, it is likely that its performance scales linearly with the number of active candidates. If pre-computation/caching, e.g., reusing past results is incorporated to overcome some of this overhead, then this deserves more discussion.
    - Related to the above point, the maximum number of samples $N_T$ ends up playing an important role in addressing the exploration-exploitation trade-off. I find that surprising as the proposed algorithm specifically aimed to offer new insights into this trade-off. Isn't it possible for the algorithm to produce running statistics indicating how much improvement can be expected if more samples are to be collected? Indeed, the ablation study with varying number of samples showed that performance reaches a plateau. Can the algorithm detect this? (e.g., in the course of arbitration as in S4.2.)
- The choice of the thresholds $\beta$ and $\rho$ seems to be absent.
- General:
    - The authors seem to use the word "optimal" in context where either it is not accurate (perhaps local optimal?, e.g., S4.3.2 optimal strategy $\omega^\ast$), properly-qualified (e.g., S2 second line), or lacking evidence (S5.2 last line). It is necessary to revise each instance and make sure it's used correctly.

**Concerns related to Neuroscience:**
- The main concern is whether the brain models mentioned are the kind of science that can undergo significant revisions in the future, and whether the inclusion of such statements is essential to an article on RL algorithms in an AI/ML venue.
- Section 2:
    - While it is beyond me to vet the contents of this section, I'm unable to take the assertions made about brain function at face value. First, I'd strongly recommend to preface this entire section by a clear statement of which articles this is drawn from. Then, it would help to qualify each assertion by specific references and statements of the form "current experimental evidence from neuroscience using brain scans/activation patterns suggests that", or "in experiments conducted on humans/animals on tasks involving", etc.
- Section 3:
    - "... designed by imitating the inference buffer of the human brain" - It would help to refer to this instead as "the XYZ model for inference in the human brain", rather than the "human brain" itself, still with a reference for XYZ. (Section 4.1 seems more inline with what I'm asking for here. Please follow this style consistently throughout.)
- Section 4:
    - "Motivated by the concurrent inference track in human PFC," same as the previous point.

**Presentation:**
- Section 3:
    - S3.2: This short section doesn't seem to offer much at this point. In particular, there's nothing about it that explains the proposed harmonized self-improving learning. (I wonder if this remainder of this section was deleted by mistake?)
- Section 4:
    - S4.3.2: I recommend replacing the number 200 with a hyperparameter, and explaining why this particular value was suitable for the experiments presented.
- Section 5:
    - Fig. 11 & 12: it would help to include the metrics mentioned in the caption within the figure itself.
    - S5.4
        - Fig.6(c): it seems NEXT+CAM actually increases the average path cost. The associated paragraph states that "CAM optimizes the path cost in high-dimensional environments."
- Appendix-A & B:
    - The paragraph at the top appears redundant as it does not include meaningful conclusions from the results. It would be better to include a sentence or two about each table, beyond what's in the table captions. Perhaps that was all deferred to Appendix-B. If that's the case, it would help to communicate this structure, though I'd recommend to keep each paragraph next to the relevant table. (Prefer to have the conclusions closer to the supporting evidence.)
    - Table 1: only few cells have non-zero stdev. A comment about that would be helpful, and I wonder if this indicates that experiment configurations need to be revised to include more challenging cases over all environments. Specifically, it would help to supplement this table with a new set of experiments (and table) designed to show HLCA failure cases, e.g., average success rate near 80%, and show how far other baselines regress for the same test cases.
    - Table 3: the gap in HLCA is highest for UR5. I wonder if this indicates different priors/hyperparameters are needed for this environment. It would help to include a comment about how this gap may be reduced by specialization of the proposed approach.
- Appendix-C:
    - Please indicate that this entire section is focused on the HCIL method, and revise figure captions to mention this as was done for Fig.9. Looking into this again, it's not immediately clear how the first paragraph+Fig.8 differs from the second paragraph+Fig.9.

**Nitpicking:**
- Abstract:
    - "meticulously crafted" appears too strong at this point, that only detracts from the main content. Recommend to replace it with simply "designed"
- Section 3:
    - S3.1: unmatched parenthesis in the definition of $U_i$.
    - S3.2: all possible histories *before* t-episodes
    - S4.2: Candidate states are sample(d) guided by
- Section 4:
    - S4.3.1: $Z_t^\mu$ and $Z_t^\lambda$ are "regularization factors" -> normalization?
    - S4.3.1: be direct as a constant -> be directly used as?
    - S4.3.2: many occurrences of "cycle" are better replaced with "iteration" or "epoch"

---

> ### Author Response · Authors · 2023-11-20
> **Response to Reviewer Api9 1/3**
>
> Thanks for the generally positive comments and constructive suggestions. We have updated manuscript accordingly to make it clearer. We address the corresponding questions below.
>
> 1. Pseudocode of the algorithm
>
> We have provided the pseudocodes of all our algorithms in Appendix E of the updated manuscript.
>
> 2. Doubts about Table 1: comparison with more challenging benchmarks
>
> Following the reviewers’ suggestion, we generated two groups of new test problems in the 2D environment, i.e., Dense2 and Large2. The environment settings are as follows: (i) Hard2: size, $15\times15$, obstacle density, no less than 0.46. (ii) Dense2: size, $15\times15$, obstacle density, no less than 0.60. (iii) Large2: size, $50\times50$, obstacle density, no less than 0.50. Maintaining consistent parameters across the board, we tested for all methods in new problems. The results are as follows and the complete table have been updated in the revised Appendix D.
>
> |Dense2|success rate|collision checks|path cost|running time|
> |:----:|:----:|:----:|:----:|:----:|
> |HLCA|0.99 $\pm$0.01|1959.04$\pm$22.88|2.56$\pm$0.02|12.18$\pm$0.55|
> |GNN+smoother|0.96$\pm$0.01|2727.04$\pm$43.06|3.25$\pm$0.02|5.65$\pm$0.12|
> |GNN|0.96$\pm$0.01|2304.78$\pm$41.51|3.96$\pm$0.02|5.45$\pm$0.12|
> |NEXT|0.82$\pm$0.01|5032.77$\pm$50.21|2.38$\pm$0.02|15.02$\pm$1.89|
> |BIT*|0.95$\pm$0.03 |2628.65$\pm$22.39|3.44$\pm$0.01|1.39$\pm$0.03|
> |LazySP|0.99$\pm$0.01|2001.21$\pm$19.35|3.44$\pm$0.01|3.54$\pm$0.11|
> |informted-RRT*|0.19$\pm$0.01|4911.51$\pm$ 20.36|1.80$\pm$0.04|1.07$\pm$0.02|
> |RRT*|0.19$\pm$0.01|4929.20$\pm$30.44|1.79$\pm$0.04|1.00$\pm$0.00|
>
> |Large2|success rate|collision checks|path cost|running time|
> |:----:|:----:|:----:|:----:|:----:|
> |HLCA|0.05$\pm$0.00|3256.82$\pm$10.25|2.52$\pm$0.04|68.53$\pm$1.26|
> |GNN+smoother|0.02$\pm$0.00|5135.97$\pm$30.76|2.82$\pm$ 0.08|378.33$\pm$0.31|
> |GNN|0.02$\pm$0.00| 5135.39$\pm$ 31.31|3.96$\pm$0.02|378.31$\pm$ 3.38|
> |NEXT|0.00$\pm$0.00|/|/|85.80$\pm$1.28|
> |BIT*|0.05$\pm$0.00|9843.90$\pm$129.08|3.22$\pm$0.47|9.80$\pm$0.22|
> |LazySP|0.02$\pm$0.00|3162.30$\pm$15.08|3.44$\pm$0.01|3.54$\pm$0.11|
> |informted-RRT*|0.69$\pm$0.01|2836.94$\pm$61.29|10.42$\pm$0.03|3.67$\pm$0.07|
> |RRT*|0.00$\pm$0.00|/|/|1.00$\pm$0.03|
>
> Compared to Hard2, operating in the Dense2 environment with a high obstacle density demands increased costs, collision checks, and runtime to reach the goal state. Our method performs best in terms of success rate and collision checks. While RRT* and informed-RRT* demonstrate a low path cost, their success rate lags behind other methods, possibly attributed to their efficacy in solving easy problems. NEXT exhibits lower path costs compared to other learning-based methods, but the collision checks are nearly twice as high.
>
> For large environments, all algorithms demonstrate a notable reduction in success rate. Only a few instances where the distance between the start state and the goal state is not long succeed in finding a path. The majority of failures may be attributed to the limitation of the maximum number of samples, set as 1000. Particularly noteworthy is the superior performance of informed-RRT*, because informed sampling limits the sampling area and reduces the need for samples. For future work, we will try to set a higher maximum number of samples and introduce informed sampling to enhance the performance of our algorithm in large environments.
>
> 3. Total running time
>
> We demonstrated the total running time for 1000 test problems across all algorithms to find a successful path. A snapshot of the results is as follows. The complete table is updated in  Appendix A of the revised manuscript. To be more clearly represented in the figure, the results are normalized based on the running time of RRT*. HLCA, GNN+smoother, GNN and NEXT are learning-based algorithms.
>
> |$ $|Easy2|Hard2|Maze3|UR5|Kuka7|Kuka14|
> |:----:|:----:|:----:|:----:|:----:|:----:|:----:|
> |HLCA |3.78 $\pm$0.04|5.86$\pm$0.11|3.64$\pm$0.13|15.18$\pm$0.45|12.13$\pm$0.53|10.78$\pm$0.42|
> |GNN|1.70$\pm$0.10|2.86$\pm$0.05|7.16 $\pm$0.42|1.98$\pm$0.32|4.85$\pm$1.86|1.11$\pm$ 0.31|
> |NEXT|3.17$\pm$0.04|5.13$\pm$0.09|2.62$\pm$0.04|23.49$\pm$0.06|18.57$\pm$1.00|15.89$\pm$0.13|
> |BIT*|0.47$\pm$0.01|0.73$\pm$0.02|2.01$\pm$0.15|1.44$\pm$0.44|2.25$\pm$1.09|0.83 $\pm$0.43|
> |LazySP|0.84$\pm$0.03|1.48$\pm$0.02|4.22$\pm$0.12|2.27$\pm$0.17|1.93$\pm$0.01|0.99$\pm$0.16|
> | RRT*|1.00$\pm$0.01|1.00$\pm$0.01|1.00$\pm$0.01|1.00$\pm$0.00|1.00$\pm$0.03|1.00$\pm$0.01|
>
> A common concern about learning-based algorithms is that their time cost due to the frequent calling of a large neural network model at the inference phase, such as NEXT. The running time is higher than non-learning algorithms because our algorithm is improved based on NEXT and runs multiple candidate strategies concurrently. Nevertheless, our algorithm reduces the running time compared with NEXT. Future work could focus on using pre-computation or caching, e.g., reusing past results to overcome some of this overhead.

---

> ### Author Response · Authors · 2023-11-20
> **Response to Reviewer Api9 2/3**
>
> 4. Ablation study about the maximum number of samples
>
> Following the reviewers’ suggestion, we varied the maximum sampling number $N_T$ among {50, 100, 200, 500, 750, 1000, 2000, 4000, 6000}  on HLCA. The results are shown in Fig.9 of the updated Appendix B. From the observation of the results, HLCA reaches a plateau in all environments when $N_T$ is 500 in success rate and path cost. For failure cases, robots must generate more samples before successfully finding a collision-free path. This implies an increased demand for collision checks and the computation cost increases on both CPU and GPU.
>
> 5. The choice of the thresholds $\alpha$ and $\beta$
>
> Theoretically, larger thresholds $\alpha$ and $\beta$ lead to better results. However, setting a higher threshold will result in longer experiment runtime, with marginal or possibly no performance improvement. During the experimental process, we try to strike a balance between efficiency and performance. $\alpha$ and $\beta$ are set as 0.7 and 0.9, respectively. Besides, we provide details about the hyperparameters of the algorithm in Appendix C of the updated manuscript.
>
> 6. Revision of "optimal"
>
> We have revised each “optimal”in the updated manuscript.
>
> 7. Concerns related to Neuroscience
>
> Truly, the brain mechanism is the kind of science that can undergo significant revisions in the future. Within the last two years, there is still work in neuroscience that supports the brain mechanism of human decision-making reasoning [1, 2]. Following the reviewers’ suggestion, we make the relevant statement in S2 of the updated manuscript. "The brain mechanism about human decision-making reasoning upon which we draw in this study is confined to the outcomes that preceded our investigation."
>
> We have clarified statements and references in S2. "The current experimental evidence from neuroscience using brain scans suggests that the concurrent reasoning mechanism in PFC includes two concurrent inference tracks [3]."
>
> Following the reviewers’ suggestion, we changed it to "...is designed by imitating the inference buffer of the concurrent reasoning model in PFC [3]." in S3. We changed it to "Motivated by concurrent inference tracks in the concurrent reasoning model  [3]" in S4.
>
> [1] Miriam C Klein-Fl ̈ugge, Alessandro Bongioanni, and Matthew FS Rushworth. Medial and orbital frontal cortex in decision-making and flexible behavior. Neuron, 2022.
>
> [2] Anna Cremer, Felix Kalbe, Jana Christina M ̈uller, Klaus Wiedemann, and Lars Schwabe. Disentangling the roles of dopamine and noradrenaline in the exploration-exploitation tradeoff during human decision-making. Neuropsychopharmacology, 8(7):1078–1086, 2023.
>
> [3] Ma ̈el Donoso, Anne GE Collins, and Etienne Koechlin. Foundations of human reasoning in the prefrontal cortex. Science, 344(6191):1481–1486, 2014.
>
> 8. Presentation about S3.2 and Fig.11, Fig.12
>
> To avoid repetition, only the relevant notations are introduced in S3.2, while the essential components of harmonized self-improving learning are introduced in S4.3. In the updated manuscript, the title of S3.2 has been modified to "Notations". We added the path cost and collision checks in the captions of Fig.11 and Fig.12.
>
> 9. Concerns about training set size
>
> Thanks and we have replaced 200 with the parameter $m$ in the updated manuscript and listed $m$ in Appendix C. The training size is also set to 200 in similar work [4, 5]. Chen et.al. [5] conducted further experiments to analyze the effect of the training size. There were 2000 training samples in their experiments. They find with 10% (200) problems of the original training set, the model is robust and obtains the best performance. Therefore, with the same number of training samples, we also set the training size to 200 in our experiments.
>
> [4] Binghong Chen, Bo Dai, Qinjie Lin, Guo Ye, Han Liu, and Le Song. Learning to plan in high dimensions via neural exploration-exploitation trees. International Conference on Learning Representations, 2020.
>
> [5] Chenning Yu and Sicun Gao. Reducing collision checking for sampling-based motion planning using graph neural networks. Advances in Neural Information Processing Systems, 34:4274–4289, 2021.
>
> 10. Doubts about Fig.6(c)
>
> Sorry for the inappropriate expression. We've removed this sentence from the updated manuscript. Despite the increase in path cost, we think that the CAM helps improve planning efficiency. Checking collisions with obstacles is vital for motion planning. Hence, we prioritized the more important collision checks in our experiments. Even though the path cost is higher than that of the NEXT in some environments, there is a significant improvement in the collision checks and success rate.
>
> 11. Presentation about Appendix A and B
>
> Following the reviewers’ suggestion, we have merged the former Appendix A and B into Appendix A in the updated manuscript. The paragraphs in the former Appendix B have been relocated beside the corresponding table.

---

> > ### Author Response · Authors · 2023-11-20
> > **Response to Reviewer Api9 3/3**
> >
> > 12. Doubts about Table 3
> >
> > Hyperparameters are kept consistent across all methods and all environments in this experiment. Unlike other robotic arm environments, obstacles are mainly concentrated around the robotic arm in UR5. There are two reasons why the gap in HLCA is highest for UR5. (i) Tree-based sampling methods such as RRT* have lower success rates due to sampling from the neighborhoods. For the failure cases, the HLCA needs to spend more path cost to be successful. (ii) We prioritize the reduction of collision checks and make some sacrifices in terms of path cost.
> >
> > Hence, it is reasonable that the gap can be narrowed through these two ways. First, the sampling scale can be extended when expanding a new state. Second, the path cost can be given a higher weight in the process of making reliability inferences.
> >
> > 13. Concerns on Table 1 in Appendix A
> >
> > In addition to the consideration that the environment may not be quiet challenging, only a few cells have non-zero stdev is also linked to the calculation of the success rate. The success rate represents the proportion of successful paths found over 1000 test problems. Although the stdev in some cells exhibits 0.00, the success paths differ by more than 40 in different randomized seed scenarios. Following the reviewers’ suggestion, we generated two groups of more challenging benchmarks. Details can be found in reply 4.
> >
> > 14. Revision of former Appendix C
> >
> > Sorry for the lack of clarity. The title of Fig.8 has been revised in the updated manuscript, accompanied by the inclusion of a paragraph elucidating the purpose of this section. "We investigate the inference of the maximum sampling number $N_T$ on CAM and HLCA, respectively."
> >
> > 15. Typos in writing
> >
> > We have improved all the listed typos in the updated manuscript.
> >
> > 16. Doubts about $Z_{t}^{\lambda}$ and $Z_{t}^{\mu}$
> >
> > $Z_{t}^{\lambda}$ and $Z_{t}^{\mu}$ denote the normalization operation.

---

> > > ### Comment · Reviewer_Api9 · 2023-11-21
> > > **Acknowledgement**
> > >
> > > Thanks for addressing my comments. I may need to adjust my scores after digesting the entirety of the rebuttal.

---

> > > > ### Author Response · Authors · 2023-11-22
> > > > **Thank you**
> > > >
> > > > We thank the reviewer for the important comments and for reading our response. We will be sure to update the paper according to the suggestions.

---

### Official Review · Reviewer_a5bK · 2023-11-01

**Soundness:** 2 fair
**Presentation:** 2 fair
**Contribution:** 1 poor
**Rating:** 3
**Confidence:** 4

**Summary:**

The manuscript proposes Harmonized Learning with Concurrent Arbitration (HLCA), a brain-inspired Deep Reinforcement Learning (DRL) algorithm for motion planning. This is inspired by the human capability for inferring concurrently and harmonizing strategies.

**Strengths:**

The algorithm sees quite novel.

**Weaknesses:**

The authors do not provide sufficient details about the parameters of the algorithm to compare with, which impairs the credibility of the experimental results. In addition, the runtime for success rates must be reported as well.

In addition, the benchmarked tasks seem fairly easy; it would be more convincing if the authors carried out experiments on larger 2D maze maps or more obstacle-rich arm motion planning problems.

There are other learning-based algorithms to compare with. The reviewer would like to see the comparison results against those learning methods: Zhang, Ruipeng, et al. "Learning-based Motion Planning in Dynamic Environments Using GNNs and Temporal Encoding." Advances in Neural Information Processing Systems 35 (2022): 30003-30015..

Minor: Section 3.1 miss left parenthesis.

**Questions:**

1. In Figure 4(a), I am curious about why success rates all drop down to 0.4 for Ur5, which are supposed to be quite easy for sampling algorithms such as RRT*.

2. The benchmarked tasks are all short-horizon problems. These tasks can be quite easy for the algorithms to compare with. The reviewer would like to see to larger 2D maze map or more obstacle-rich arm motion planning problems.

---

> ### Author Response · Authors · 2023-11-20
> **Response to Reviewer a5bK 1/2**
>
> Thanks for your careful review and important suggestions for the comparison with a learning-based approache and more challenging environments. We have revised manuscript accordingly to make it clearer. We address the specific questions as follows.
>
> 1. Parameters
>
> We have provided details about the parameters of the algorithm in Appendix C of the revised manuscript. These parameters  include learning rates, epochs, reliability thresholds $\alpha$, $\beta$, and so on.
>
> 2. Total running time
>
> We demonstrated the total running time for 1000 test problems across all algorithms to find a successful path. A snapshot of the results is as follows. The complete table is updated in  Appendix A of the revised manuscript. To be more clearly represented in the figure, the results are normalized based on the running time of RRT*. HLCA, GNN+smoother, GNN, and NEXT are learning-based algorithms, others are non-learning algorithms.
>
> |$ $|Easy2|Hard2|Maze3|UR5|Kuka7|Kuka14|
> |:----:|:----:|:----:|:----:|:----:|:----:|:----:|
> |HLCA |3.78 $\pm$0.04|5.86$\pm$0.11|3.64$\pm$0.13|15.18$\pm$0.45|12.13$\pm$0.53|10.78$\pm$0.42|
> |GNN|1.70$\pm$0.10|2.86$\pm$0.05|7.16 $\pm$0.42|1.98$\pm$0.32|4.85$\pm$1.86|1.11$\pm$ 0.31|
> |NEXT|3.17$\pm$0.04|5.13$\pm$0.09|2.62$\pm$0.04|23.49$\pm$0.06|18.57$\pm$1.00|15.89$\pm$0.13|
> |BIT*|0.47$\pm$0.01|0.73$\pm$0.02|2.01$\pm$0.15|1.44$\pm$0.44|2.25$\pm$1.09|0.83 $\pm$0.43|
> |LazySP|0.84$\pm$0.03|1.48$\pm$0.02|4.22$\pm$0.12|2.27$\pm$0.17|1.93$\pm$0.01|0.99$\pm$0.16|
> | RRT*|1.00$\pm$0.01|1.00$\pm$0.01|1.00$\pm$0.01|1.00$\pm$0.00|1.00$\pm$0.03|1.00$\pm$0.01|
>
> A common concern about learning-based algorithms is that their time cost due to the frequent calling of a large neural network model at the inference phase, such as NEXT. The running time is higher than non-learning algorithms because our algorithm is improved based on NEXT and runs multiple candidate strategies concurrently. Nevertheless, our algorithm reduces the running time compared with NEXT. Future work could focus on using pre-computation or caching, e.g., reusing past results to overcome some of this overhead.
>
> 3. Comparison with more challenging environments
>
> Following the reviewers’ suggestion, we generated two groups of new test problems in the 2D environment, i.e., Dense2 and Large2. The environment settings are as follows: (i) Hard2: size, $15\times 15$, obstacle density, no less than 0.46. (ii) Dense2: size, $15\times 15$, obstacle density, no less than 0.60. (iii) Large2: size, $50\times 50$, obstacle density, no less than 0.50. Maintaining consistent parameters across the board, we tested for all methods in new problems. The results are as follows and the complete table have been updated in the revised Appendix D.
>
> |Dense2|success rate|collision checks|path cost|running time|
> |:----:|:----:|:----:|:----:| :----:|
> |HLCA|0.99 $\pm$0.01|1959.04$\pm$22.88|2.56$\pm$0.02|12.18$\pm$0.55|
> |GNN+smoother|0.96$\pm$0.01|2727.04$\pm$43.06|3.25$\pm$0.02|5.65$\pm$0.12|
> |GNN|0.96$\pm$0.01|2304.78$\pm$41.51|3.96$\pm$0.02|5.45$\pm$0.12|
> |NEXT|0.82$\pm$0.01|5032.77$\pm$50.21|2.38$\pm$0.02|15.02$\pm$1.89|
> |BIT*|0.95$\pm$0.03 |2628.65$\pm$22.39|3.44$\pm$0.01|1.39$\pm$0.03|
> |LazySP|0.99$\pm$0.01|2001.21$\pm$19.35|3.44$\pm$0.01|3.54$\pm$0.11|
> |informted-RRT*|0.19$\pm$0.01|4911.51$\pm$ 20.36|1.80$\pm$0.04|1.07$\pm$0.02|
> |RRT*|0.19$\pm$0.01|4929.20$\pm$30.44|1.79$\pm$0.04|1.00$\pm$0.00|
>
> |Large2|success rate|collision checks|path cost|running time|
> |:----:|:----:|:----:|:----:|:----:|
> |HLCA|0.05$\pm$0.00|3256.82$\pm$10.25|2.52$\pm$0.04|68.53$\pm$1.26|
> |GNN+smoother|0.02$\pm$0.00|5135.97$\pm$30.76|2.82$\pm$ 0.08|378.33$\pm$0.31|
> |GNN|0.02$\pm$0.00| 5135.39$\pm$ 31.31|3.96$\pm$0.02|378.31$\pm$ 3.38|
> |NEXT|0.00$\pm$0.00|/|/|85.80$\pm$1.28|
> |BIT*|0.05$\pm$0.00|9843.90$\pm$129.08|3.22$\pm$0.47|9.80$\pm$0.22|
> |LazySP|0.02$\pm$0.00|3162.30$\pm$15.08|3.44$\pm$0.01|3.54$\pm$0.11|
> |informted-RRT*|0.69$\pm$0.01|2836.94$\pm$61.29|10.42$\pm$0.03|3.67$\pm$0.07|
> |RRT*|0.00$\pm$0.00|/|/|1.00$\pm$0.03|
>
> Compared to Hard2, operating in the Dense2 environment with a high obstacle density demands increased costs, collision checks, and runtime to reach the goal state. Our method performs best in terms of success rate and collision checks. While RRT* and informed-RRT* demonstrate a low path cost, their success rate lags behind other methods, possibly attributed to their efficacy in solving easy problems. NEXT exhibits lower path costs compared to other learning-based methods, but the collision checks are nearly twice as high.

---

> ### Author Response · Authors · 2023-11-20
> **Response to Reviewer a5bK 2/2**
>
> For large environments, all algorithms demonstrate a notable reduction in success rate. Only a few instances where the distance between the start state and the goal state is not long succeed in finding a path. The majority of failures may be attributed to the limitation of the maximum number of samples, set as 1000. Particularly noteworthy is the superior performance of informed-RRT*, because informed sampling limits the sampling area and reduces the need for samples. For future work, we will try to set a higher maximum number of samples and introduce informed sampling to enhance the performance of our algorithm in large environments.
>
> 4. Comparison with GNN-TE
>
> Following the reviewers’ suggestion, we have implemented GNN-TE[1] as another baseline method to compare with. A snapshot of the results is as follows, and we will update the manuscript with the complete tables.
>
> Average collision checks.
>
>  |$ $|Easy2|Hard2|Kuka7| Kuka13|Kuka14|
>  |:----:|:----:|:----:|:----:|:----:|:----:|
> |HLCA |129.05$\pm$ 3.04|342.35$\pm$ 13.90|479.86$\pm$ 63.64|500.43$\pm$ 269.34|508.03$\pm$52.56|
> |GNN-TE|412.66$\pm$ 8.16|659.90$\pm$ 13.52|537.08$\pm$ 61.64|498.25$\pm$56.89|497.20$\pm$ 89.22|
> |GNN | 416.54$\pm$ 9.23|720.47$\pm$ 7.46 |633.79$\pm$ 60.39|522.87$\pm$64.25|564.92$\pm$ 100.08|
> |NEXT|272.28$\pm$ 14.81|1135.6$\pm$ 37.7 |2516.75 $\pm$ 45.73|4870.57$\pm$ 129.12|4557.94 $\pm$ 71.22|
>
> Average path cost.
>
>  |$ $|Easy2|Hard2|Kuka7| Kuka13|Kuka14|
>  |:----:|:----:|:----:|:----:|:----:|:----:|
> |HLCA |1.08$\pm$ 0.00|2.01$\pm$ 0.00|6.30$\pm$ 0.03|9.67$\pm$ 0.04|9.78$\pm$ 0.09|
> |GNN-TE|1.51$\pm$ 0.00|2.80$\pm$ 0.01|8.96$\pm$ 0.11|16.88$\pm$ 0.12|16.69$\pm$ 0.22|
> |GNN | 1.50$\pm$ 0.00|2.89$\pm$ 0.01|9.15$\pm$ 0.07|17.19$\pm$ 0.23|16.83$\pm$ 0.26|
> |NEXT|1.06$\pm$ 0.00|2.04$\pm$ 0.00|7.90$\pm$ 0.05|10.25$\pm$ 0.11|10.73$\pm$ 0.09|
>
> After a careful reading, it is noted that the GNN-TE is particularly designed for dynamic environments. With full knowledge of the obstacles, GNN-TE leverages graph neural networks and temporal encoding framework to achieve motion planning problems in dynamic environments. However, our work mainly concentrates on static environments, in which the GNN-TE obtains approximate performance of the GNN. It is expected that GNN-TE will perform better in dynamic environments, which would be our future study topic.
>
> [1] Zhang, Ruipeng, et al. "Learning-based Motion Planning in Dynamic Environments Using GNNs and Temporal Encoding." Advances in Neural Information Processing Systems 35 (2022): 30003-30015.
>
> 5. Doubts about UR5
>
> There are two reasons why this happens. Unlike other robotic arm environments, obstacles are mainly concentrated around the robotic arm in UR5. On the one hand, RRT* samples a state from free space, and determines a new state to connect to through the steer function. The steer function only generates states that are close to the search tree within neighboring areas. Therefore, in environments with high obstacle density, tree-based sampling methods such as RRT* struggle to expand new states. On the other hand, the maximum sample number is limited to 1000 in our experiments. Under this condition, RRT* reaches the maximum budget of the sampling nodes and is forced to be terminated without finding a solution. Consequently, it directly results in a substantial decrease in the success rate of RRT*.
>
> 6. Minor comments
>
> We have improved the typo mentioned in minor comments in the revised manuscript.

---

### Meta-Review · Area_Chair_S3fZ · 2023-12-09

**Metareview:**

This paper proposes a neuroscience-inspired approach for motion planning (HLCA, Harmonized Learning with Concurrent Abritration). At its core, the paper proposes two modules:

(1) CAM (Concurrent Arbitration Module) that balances exploration and exploitation. States are sampled as in an infinite-armed bandit problem. Average reward and standard deviation are computed via kernel smoothing, and the balance betwen exploration and exploitation changes over time, as more measurements are taken.

(2) HSIL (Harmonized Self-Improving Learning) which learns a policy and a value function from Monte Carlo rollouts and evaluations.

The paper evaluates the planner on a number of robotic maze navigation and manipulation tasks and compares it against well-known planning baselines (eg BIT*). Overall, results suggest that the learned planner requires fewer collision checks than alternatives, but spends more time planning than state of the art planners. As such, it is not clear what the advantage of using this method would be.

I think the paper needs to be improved in the following directions:

(a) The terminology is quite confusing. I am not letting this affect my evaluation very much, but it is worth re-writing significant parts of the paper using standard RL terminology. I found terms such as HSIL and CAM to be obfuscating the meanings behind the method, rather than helping understand the method.

(b) Improve the runtime, or rather the number of nodes that are opened in the tree to be competitive with BIT*. Roboticists will not use this approach if it does not produce good plans in an efficient way.

(c) The claim about CAM not relying on the UCB rule because "As the algorithm proceeds, the number of
states gradually increases and their values become correlated, the traditional UCB algorithm is not
directly applicable to balance exploration and exploitation.". Can you provide a reference for this? Also, what type of correlations are you referring to?

I think the paper needs one more iteration fo work before it is ready for publication, but I encourage the authors to keep pursuing this interesting direction.

**Justification For Why Not Higher Score:**

See reasons a-c above.

**Justification For Why Not Lower Score:**

N/A

---

### Decision · Program_Chairs · 2024-01-16

Reject